# Rapid protection induced by a single-shot Lassa vaccine in male cynomolgus monkeys

Mathieu Mateo [1,2], Stéphanie Reynard [1,2], Natalia Pietrosemoli[3], Emeline Perthame [3], Alexandra Journeaux[1,2], Kodie Noy[1,2], Clara Germain[1,2], Xavier Carnec [1,2], Caroline Picard [1,2], Virginie Borges-Cardoso[1,2], Jimmy Hortion[1,2], Hélène Lopez-Maestre [3], Pierrick Regnard[4], Lyne Fellmann[4], Audrey Vallve[5], Stéphane Barron[5], Ophélie Jourjon[5], Orianne Lacroix[5], Aurélie Duthey[5], Manon Dirheimer[6], Maïlys Daniau[7], Catherine Legras-Lachuer[7], Caroline Carbonnelle[5], Hervé Raoul [5], Frédéric Tangy [8] & Sylvain Baize [1,2] ✉

Lassa fever hits West African countries annually in the absence of licensed vaccine to limit the burden of this viral hemorrhagic fever. We previously developed MeV-NP, a single-shot vaccine protecting cynomolgus monkeys against divergent strains one month or more than a year before Lassa virus infection. Given the limited dissemination area during outbreaks and the risk of nosocomial transmission, a vaccine inducing rapid protection could be useful to protect exposed people during outbreaks in the absence of preventive vaccination. Here, we test whether the time to protection can be reduced after immunization by challenging measles virus pre-immune male cynomolgus monkeys sixteen or eight days after a single shot of MeV-NP. None of the immunized monkeys develop disease and they rapidly control viral replication. Animals immunized eight days before the challenge are the best controllers, producing a strong CD8 T-cell response against the viral glycoprotein. A group of animals was also vaccinated one hour after the challenge, but was not protected and succumbed to the disease as the control animals. This study demonstrates that MeV-NP can induce a rapid protective immune response against Lassa fever in the presence of MeV pre-existing immunity but can likely not be used as therapeutic vaccine.

Lassa fever is a major public health issue in West Africa, causing thousands of deaths each year. The disease is difficult to diagnose due to the low specificity of early symptoms, such as fever, fatigue, and headache[1]. Thus, most cases are laboratory-confirmed at an advanced stage of the disease and the overall case fatality rate among hospitalized patients ranges from 15 to 30%[2]. Supportive care and symptomatic treatment appear to improve survival[3], whereas the

benefit of ribavirin on survival has been seriously questioned[4]. Hence, there is an urgent need for the development of treatments and vaccines.

The etiological agent of Lassa fever is Lassa virus (LASV), an arenavirus that circulates naturally in rodents of the *Mastomys* genera[5,6]. Rodents are the main source of human infection[7], which generally occurs after contact with infectious fluids or the inhalation of

[1]Unité de Biologie des Infections Virales Emergentes, Institut Pasteur, 69007 Lyon, France. [2]Centre International de Recherche en Infectiologie (CIRI), Université de Lyon, INSERM U1111, Ecole Normale Supérieure de Lyon, Université Lyon 1, CNRS UMR5308, 69007 Lyon, France. [3]Institut Pasteur, Université Paris Cité, Bioinformatics and Biostatistics Hub, Paris, France. [4]SILABE, Université de Strasbourg, fort Foch, Niederhausbergen, France. [5]Laboratoire P4 INSERM – Jean Mérieux, INSERM US003, 69007 Lyon, France. [6]INSERM, Délégation Régionale Auvergne Rhône-Alpes, 69500 Bron, France. [7]Viroscan 3D SAS, Trévoux, France. [8]Vaccine Innovation Laboratory, Institut Pasteur, 75015 Paris, France. ✉e-mail: sylvain.baize@pasteur.fr

contaminated dust. Annual peaks of epidemics are often observed during the dry season, between January and March, when the reservoir population is increasing and comes into closer proximity to human activities to find resources[8]. Nigeria, one of the top countries in terms of LASV surveillance, reports several hundred LASV cases annually, with important geographical clusters, notably, in the Ondo and Edo states[9]. In addition, clusters of nosocomial Lassa fever have been observed in the past and still occur, despite the improvement of patient management care in West African countries[10,11].

The World Health Organization (WHO) has ranked Lassa fever in the top priority list of viral diseases requiring urgent vaccine development and they published a target product profile (TPP) for a Lassa fever vaccine[12]. According to this TPP, the vaccine should be safe for all age groups, demonstrate at least 90% efficacy in preventing infection or disease, confer long-lasting immunity for several years, preferentially after a single immunization, and cross-protect against several LASV lineages. Although a preventive vaccine is preferred over an emergency vaccine for mass vaccination, an emergency vaccine would be highly valuable in stopping the spread of the disease in communities or in avoiding nosocomial outbreaks. Indeed, cluster-targeted vaccination of exposed populations could perhaps significantly reduce the number of Lassa fever cases.

Strong research efforts have recently focused on the development of Lassa fever vaccines with the support of the Coalition for Epidemic Preparedness Innovations (CEPI). Several LASV vaccine candidates have been developed over the past decades[13] but only very few of them have demonstrated efficacy in relevant nonhuman primate (NHP) models. The first LASV vaccine that has shown efficacy in NHP was a recombinant vaccinia virus expressing the LASV glycoprotein (GP) alone or in combination with the LASV nucleoprotein (NP)[14,15] but the development of this vaccine was then discontinued due to its high reactogenicity. A few years later, a recombinant vesicular stomatitis virus (VSV) expressing LASV GP (rVSVΔG-LASV-GPC) was successfully tested in NHPs against the homologous LASV strain[16]. Since then, rVSVΔG-LASV-GPC was shown to be efficacious against two other strains from lineage IV[17] and a recent study demonstrated its short-term efficacy against a strain from lineage II[18]. Another modified VSV-based platform expressing LASV GP, Vesiculovax, protected NHPs against a LASV strain from lineage II[19]. Other LASV vaccines tested successfully in NHPs against the homologous LASV strain include a Mopeia-Lassa reassortant, ML29[20], and a DNA vaccine, pLASV-GPC[21]. Among all these vaccines, several of them have entered phase I clinical trials: rVSVΔG-LASV-GPC (NCT04794218), Vesiculovax (PACTR202108781239363), and pLASV-GPC (NCT03805984), as well as the vaccine candidate discussed here, which entered clinical development under the name MV-LASV (NCT04055454).

We originally developed this recombinant measles virus (MeV) that expresses LASV GP and NP of the prototypic Josiah strain under the name of MeV-NP. To improve the immunogenicity of the measles backbone, the vaccine was further engineered to abolish the IFN-antagonist activity of NP[22,23]. We recently demonstrated that a single shot of MeV-NP protects cynomolgus monkeys against Lassa fever, confers cross-protection against strains from distant lineages II and VII, and induces long-term immunity[23,24], thus demonstrating the potential of this candidate to address the urgent medical need presented by Lassa fever in West Africa. One concern about the use of MeV-NP in the human population is the pre-existing immunity against MeV which could affect the efficacy of the vaccine. In this study, we continued to explore the efficacy of MeV-NP against LASV by evaluating the kinetics of vaccine-induced protection, in presence of MeV pre-existing immunity. We found that the MeV-NP vaccine fully protects MeV pre-immune cynomolgus monkeys when given 16 or even 8 days before a lethal infection, but does not demonstrate therapeutic efficacy in animals vaccinated after challenge.

## Results

### A single shot of MeV-NP induces rapid protection in cynomolgus monkeys

Twelve monkeys were separated into four groups of three monkeys each: D-16, D-8, H + 1, and CT (Fig. 1A). Groups D-16 and D-8 were immunized by intramuscular (i.m.) injection of $2 \times 10^6$ TCID$_{50}$ of MeV-NP 16 or 8 days before challenge, respectively. Group H + 1 was immunized with the same dose by the same route one hour after the challenge. The CT group was not immunized. Importantly, all animals were pre-immune to MeV as they received two shots of live attenuated Measles vaccine (Serum Institute of India) at the age of two and three months. On the day of challenge (day 0), all animals received a subcutaneous injection of 1,500 FFU of LASV Josiah (Fig. 1A). The animals were then monitored for up to 28 days and sampling was performed regularly throughout the protocol to evaluate various parameters. Clinical scores were attributed to each animal based on their body temperature, body weight, dehydration, bleeding, petechia, stool aspect, and reactivity (Supplementary. Table 1). A score of 15 or above necessitated euthanasia of the animal to limit suffering. Animals were also euthanized if they reached any of the following limit points: body temperature below 35.8 °C, post anesthesia coma lasting longer than two-and-a-half hours, convulsions, or serious balance issues. On day 6, all animals showed reduced activity but low clinical scores, below 3 (Fig. 1B). In the CT and H + 1 groups, the clinical scores increased by day 9 for all animals. Animal CT_3 showed balance issues and shivering and was euthanized on day 9 with a rectal temperature of 35.6 °C and a score of 13 (Fig. 1B). On day 11, animal CT_1 also showed serious balance issues and was euthanized with a score of 19 (Fig. 1B). On day 14, animal CT_2 also showed balance issues and epistaxis and was euthanized after anesthesia with a score of 15 (Fig. 1B). In the H + 1 group, animal H + 1_1 was found moribund at day 13 and was euthanized with a score of 24 (Fig. 1B). Animals H + 1_2 and H + 1_3 presented epistaxis by day 11 and were euthanized at day 15 with low body temperatures and clinical scores of 16 and 19, respectively (Fig. 1B). In the D-16 and D-8 groups, the animals showed normal activity after day 6 and their clinical scores remained low until the end of the experiment (Fig. 1B).

In addition to rectal measures of body temperature on each sampling day, we followed the evolution of the body temperature using intraperitoneal devices. These data were collected and analyzed at the end of the experiment (Fig. 1C). The body temperature of most infected animals increased from approximately 38 °C to 41 °C between days 3 and 6. For the CT and H + 1 animals, the fever persisted for several days and the body temperature then dropped sharply in the hours preceding euthanasia (Fig. 1C). The body temperature of two D-16 animals remained elevated until day 9 and decreased to normal values by day 12 (Fig. 1C). In this group, animal D-16_2 remained normothermic for the entire experiment (Fig. 1C). The D-8 animals showed the smallest change in body temperature. Animal D-8_1 had a fever between days 3 and 9 (Fig. 1C) and animal D-8_3, a transient fever, between days 4 and 6, below 40 °C (Fig. 1C). Animal D-8_2 remained normothermic for the entire experiment (Fig. 1C).

We also measured several biochemical parameters throughout the protocol. Plasma concentrations of alanine (ALT) and aspartate (AST) aminotransferases, two markers of severe Lassa fever in both humans and monkeys[25,26], increased from day 6 or day 9 until the time of death in all CT and H + 1 animals (Fig. 1D). The plasma concentrations of C-reactive protein (CRP), a marker of inflammation, also increased continuously in CT and H + 1 animals from day 3 to the time of death (Fig. 1D). On the contrary, plasma concentrations of ALT and AST did not increase in all D-16 and D-8 animals throughout the experiment (Fig. 1D). In these groups, CRP concentrations only increased transiently between days 6 and 9, except for animals D-16_2 and D-8_2, for which no variation in CRP concentration was observed (Fig. 1D, lower center and lower right panels, green lines). The blood

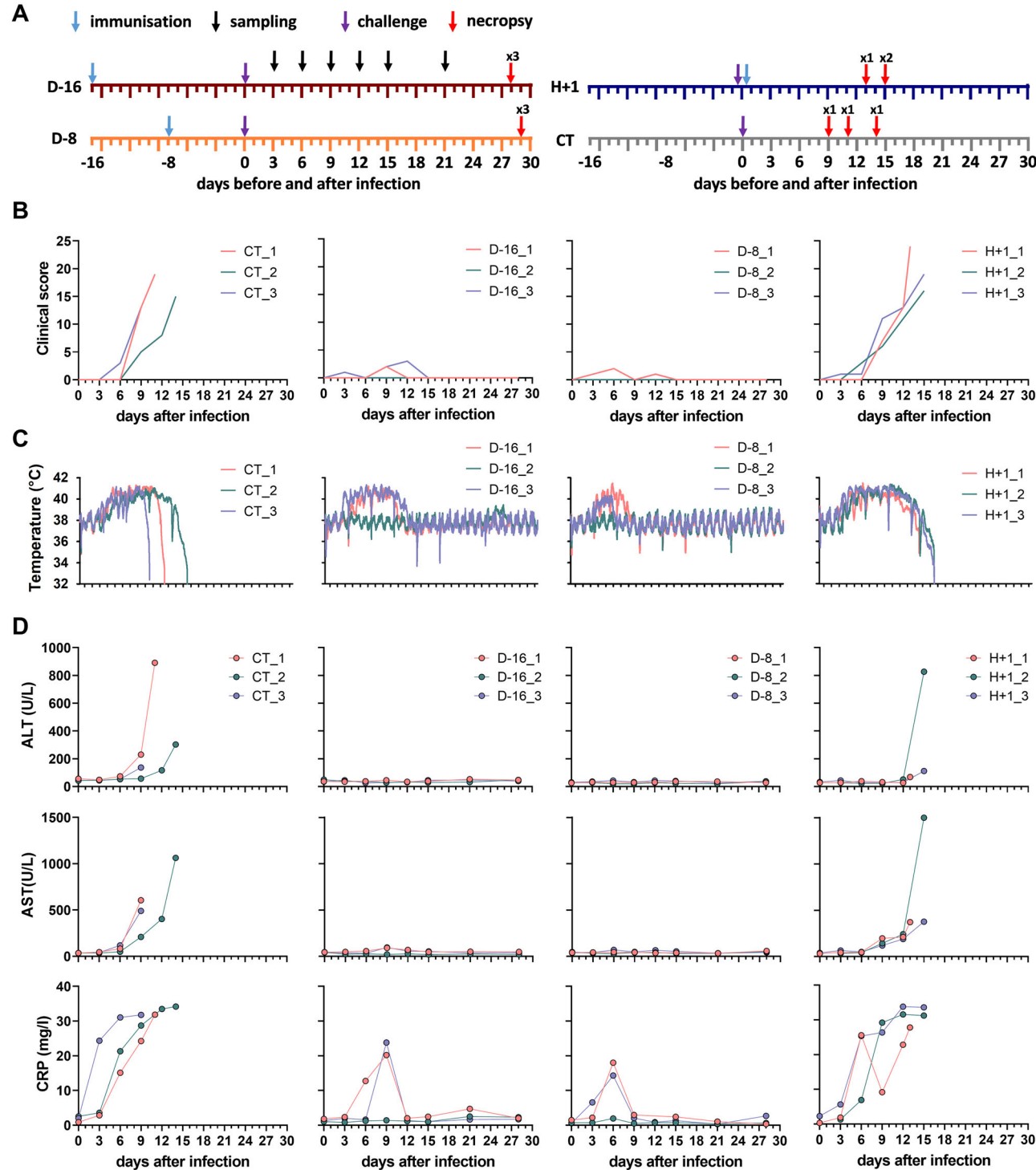

**Fig. 1 | Short-term immunization of cynomolgus monkeys with MeV-NP protects against Lassa fever. A** Outline of the experiment. Scheme presenting the three animal groups with MeV-NP immunizations indicated by blue arrows, LASV challenge by purple arrows, sampling by black arrows, and necropsy by red arrows, with the number of necropsies indicated above the arrows. D-16: animals immunized with MeV-NP 16 days before challenge. D-8: animals immunized with MeV-NP 8 days before challenge. H + 1: animals immunized with MeV-NP 1 h after challenge.

CT: unvaccinated control animals. **B** Evolution of the individual clinical scores in animals over time after the LASV infection. **C** Monitoring of body temperature by real-time measurement during the course of LASV infection. **D** Analysis of ALT, AST, and CRP plasma concentrations during the course of LASV infection. For **B**–**D**, individual data are presented for each monkey. Source data are provided as a Source Data file.

biochemistry also indicated increased plasma concentrations of lactate dehydrogenase (LDH) and urea and a drop in the albumin (ALB) concentration in the CT and H + 1 animals, whereas the D-16 and D-8 vaccinated animals did not show strong variation in the concentrations of these markers (Sup. Figure 1).

Knowing that strong inflammatory responses are also markers of severe Lassa fever in cynomolgus monkeys[23,25,27], we performed a multiplex analysis of several soluble mediators of inflammation on the samples collected during the experiment. Plasma concentrations of the pro-inflammatory cytokines IL-6 and TNF-α, as well as soluble

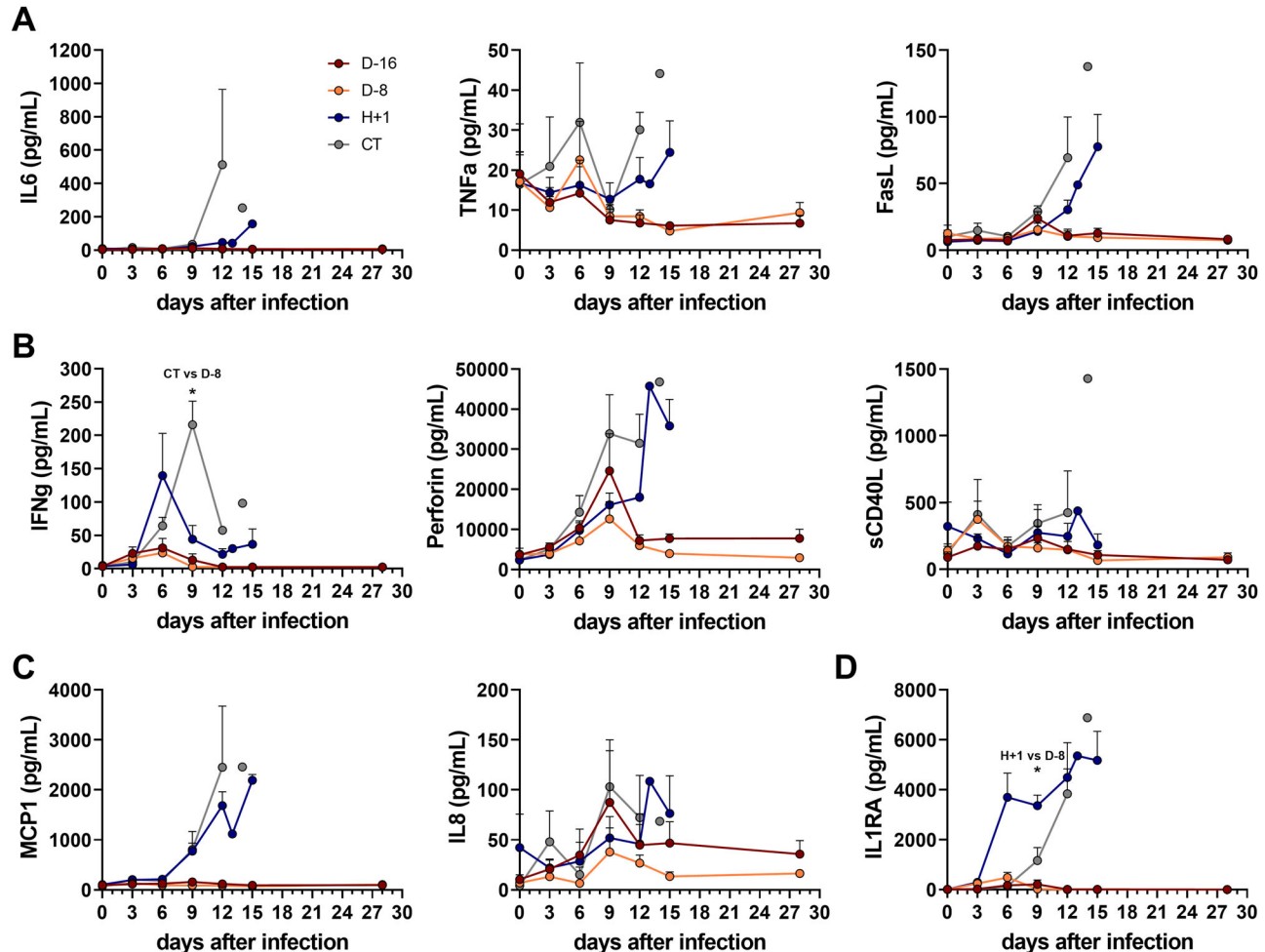

**Fig. 2 | Release of soluble mediators into the plasma of infected monkeys after LASV challenge.** Plasma concentrations of **A** the pro-inflammatory cytokines IL6, TNFα, and FasL, **B** the T-cell response-related mediators IFNγ (CT vs D-8 $p = 0.0327$), perforin, and sCD40L, **C** the chemokines MCP1 and IL8, and **D** anti-inflammatory IL1RA (H + 1 vs D-8 $p = 0.0327$). All plasma concentrations were measured by Luminex assay. Each point represents the mean ± SEM of three samples. A Kruskal–Wallis test was used after testing for normality with a Shapiro-Wilk normality test. Statistical significance: $*p \leq 0.05$. Source data are provided as a Source Data file.

FasL were all increased in the days preceding death in the CT and H + 1 animals but remained unchanged in the D-16 and D-8 animals, except for a transient increase in TNF-α concentrations on day 6 in the D-8 animals and, to a lesser extent, D-16 animals (Fig. 2A). The plasma concentrations of IFN-γ also increased intensely in the CT animals from day 3 to day 9 and in the H + 1 animals at day 6, while there was no significant change in the D-16 or D-8 animals (Fig. 2B, left panel). Perforin was also released into the plasma of the CT and H + 1 animals at high concentrations at the time of death, whereas the D-16 and D-8 animals showed only a transient increase in the perforin concentration at day 9, which was greater in the D-16 than D-8 animals. The plasma concentration of sCD40L was also increased at day 14 in animal CT_2. High concentrations of the chemokine MCP-1 were also measured late during infection only in the CT and H + 1 animals, whereas the concentration of IL-8 peaked on day 9 for the D-16, D-8, and CT animals, particularly for the CT and D-16 animals, before dropping in the following days (Fig. 2C). The plasma concentration of the anti-inflammatory cytokine IL1RA also increased dramatically in the CT and H + 1 animals only, in the days preceding their death (Fig. 2D). Therefore, D-16 or D-8 immunizations were able to prevent the strong inflammatory responses observed in non-protected animals.

Thus, all MeV-NP pre-immunized animals survived the LASV challenge without presenting overt clinical signs or markers of Lassa

fever, suggesting efficient control of LASV replication, particularly in the D-8 group. However, therapeutic vaccination failed to protect animals and did not improve survival compared to unvaccinated animals.

## A single shot of MeV-NP allows efficient control of LASV replication very early after infection

We measured the LASV infectious titers (Fig. 3A) and RNA viral loads (Fig. 3B) in plasma samples collected at various time points after challenge. In the plasma of CT and H + 1 animals, we observed increasing infectious titers from day 6 until death (Fig. 3A). In contrast, we did not detect any infectious virus at any time points after challenge in the D-16 or D-8 animals (Fig. 3A). LASV RNA was detected earlier, at day 3, in all CT and H + 1 animals and the viral loads increased continuously until the time of death, reaching up to $5.4 \times 10^9$ RNA copies/ml for animal CT_2 (Fig. 3B). Viral loads increased until day 6 or day 9 in animals D-16_1 and D-16_3, reaching between $10^6$ and $10^7$ RNA copies/mL, but no viral RNA was detected in D-16_2 (Fig. 3B). Among the D-8 animals, D-8_3 was the only one to show a low viral load at day 3, below $10^5$ RNA copies/mL (Fig. 3B).

We also transiently detected infectious titers in the nasal swabs of CT and H + 1 animals between days 6 and 15 (Supplementary Fig. 2A), but not in the nasal swabs of the D-16 or D-8 animals (Supplementary Fig. 2A). CT_3 and H + 1_2 were the only animals with

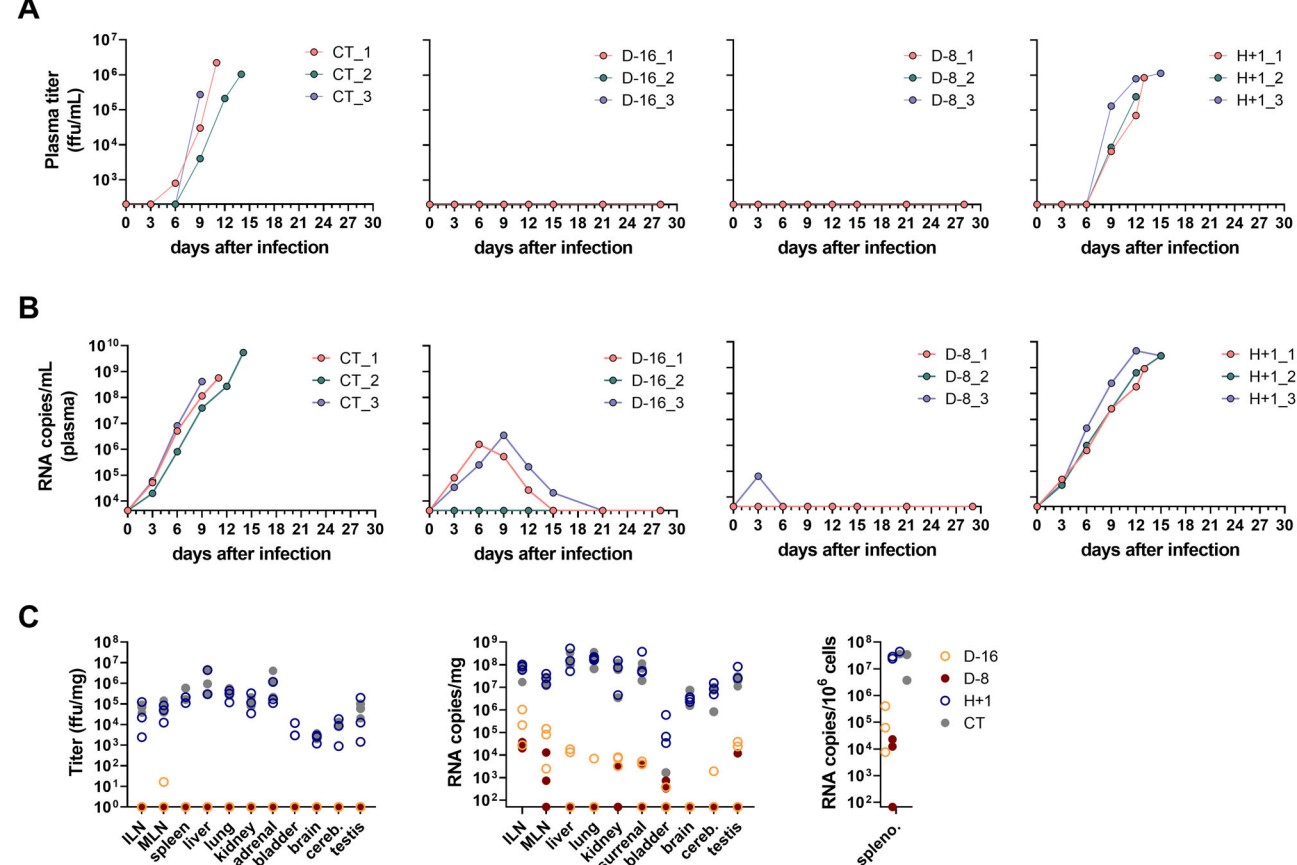

**Fig. 3 | LASV replication in plasma and organs of cynomolgus monkeys after LASV challenge. A** Quantification of infectious LASV titers (in FFU per milliliter) and **B** quantification of LASV viral loads (in RNA copies/mL), measured in plasma samples collected over the course of LASV infection from unvaccinated control animals (CT), animals vaccinated 16 days (D-16) or 8 days (D-8) before the LASV challenge, or animals vaccinated 1 hour after the LASV challenge (H + 1). Individual data are presented for each animal. **C** Quantification of infectious LASV titers and viral loads in organs collected at the time of necropsy. ILN: inguinal lymph node, MLN, mesenteric lymph node, cereb.: cerebellum, spleno.: splenocytes. Source data are provided as a Source Data file.

detectable infectious titers in the oral swab, between days 6 and 9 and at day 12, respectively (Supplementary Fig. 2A). All CT and H + 1 animals showed detectable viral loads in their swabs between days 6 and 15 (Supplementary Fig. 2B). The D-16 animals also showed low viral loads in their swabs between days 6 and 15 (Supplementary Fig. 2B), except for animal D-16_2, which completely controlled viral replication (Supplementary Fig. 2B). Animals D-8_2 and D-8_3 did not show viral loads in their swabs and animal D-8_1 had very low detectable amounts of viral RNA in its oral swab at days 9 and 12.

We collected several organs at the time of necropsy and measured the amount of infectious LASV particles (Fig. 3C, left panel) and the number of copies of LASV RNA (Fig. 3C, middle and right panels) in each. In CT and H + 1 animals, we detected infectious titers (Fig. 3C, left panel) and viral RNA (Fig. 3C, middle panel) in the inguinal and mesenteric lymph nodes, spleen, liver, lungs, kidneys, adrenals, brain, cerebellum, and testis. The bladder was the only collected organ for which we did not find any associated infectious titers in CT animals, but we detected viral RNA in the bladder of CT_2, albeit in low amounts (Fig. 3C, middle panel). Both infectious titers and viral RNA were detected in the bladder of H + 1 animals. Among the other animals, we only detected a low viral titer (16.5 focus-forming units (FFU)/mg) in the mesenteric lymph nodes of animal D-16_1. Otherwise, we did not detect any infectious titers in any organs of the other D-16 or any of the D-8 animals, despite the presence of low amounts of viral RNA in the organs of at least one surviving animal, except the brain (Fig. 3C, middle and right panels).

## Short-term vaccination allows the establishment of humoral immunity in the presence of pre-existing anti-measles immunity

To get further insights into the immune responses developed by the different animal groups after the challenge, we first measured the titers of immunoglobulin M (IgM) and G (IgG) directed against LASV antigens produced in the days following the LASV challenge (Fig. 4). A low titer of LASV-specific IgM was detected from day 0 to day 6 in D-8 animals (Fig. 4A, orange line), but not in D-16, H + 1 and CT animals. The LASV IgM titers increased rapidly between day 6 and day 9 in all animals to reach a maximum titer by day 9 (Fig. 4A). The IgM titer then remained stable for the D-16 animals (Fig. 4A, red line) and diminished slowly for the D-8 animals (Fig. 4A, orange line) until the end of the experiment. The IgG response against whole LASV antigens also increased rapidly between days 3 and 9 in the D-16 and D-8 animals (Fig. 4B, red and orange lines, respectively) but was only detected at the time of death in the CT and H + 1 animals (Fig. 4B, gray line). To gain more specificity and sensitivity in the detection of LASV-specific IgG, we performed ELISAs using recombinant NP or GP proteins. IgG titers against NP could be detected as soon as day 3 in the D-8 animals (Fig. 4C, orange line) and day 6 in the D-16 animals (Fig. 4C, red line) and were maximal on day 9. In the CT and H + 1 animals, we only detected NP-specific IgG by day 9 (Fig. 4C, gray line). Using a recombinant GP protein, we were able to detect LASV GP-specific IgG at day 0 in the D-16 animals (Fig. 4D, red line) and day 3 in the D-8 animals (Fig. 4D, orange line). Then, the GP IgG titers increased rapidly in these two groups to reach a maximum titer by day 12. In the CT and H + 1 animals, GP-specific IgG only appeared late, after day 12 (Fig. 4D, gray

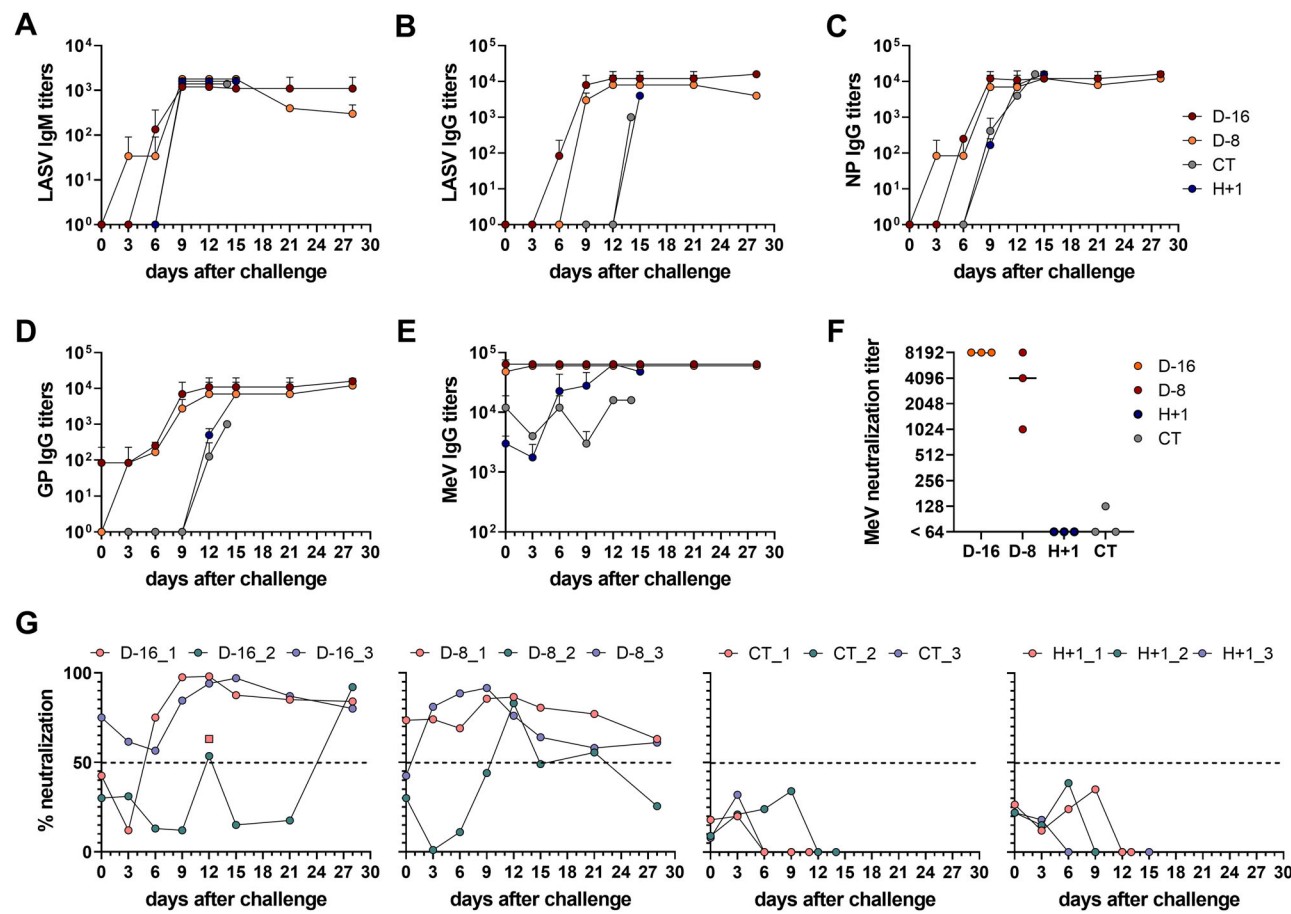

**Fig. 4 | Antibody responses induced after LASV challenge.** Detection of **A** LASV-specific IgM, **B** LASV-specific IgG, **C** NP-specific IgG, **D** GP-specific IgG, and **E** Measles-specific IgG in animals after the LASV infection by ELISA. Each point represents the mean ± SEM of three samples, except for the CT animals on day 12 (2 samples) and day 14 (one sample). For statistical purposes, the sample of animal CT_2 collected at the time of death on day 11 is represented as being on day 12, and the sample of animal H + 1_1 collected at the time of death on day 13 is represented as being on day 15. A Kruskal–Wallis test was used after testing for normality with a Shapiro-Wilk normality test. **F** Measles virus neutralizing antibody titers, expressed as reciprocal of the highest dilution blocking infection of Vero NK cells by 250 plaque-forming units (pfu) of MeV-GFP. The bars represent the mean of $n = 3$ independent samples. **G** Percentage of seroneutralization measured by plaque reduction neutralization assay with 1/40 dilutions of plasma collected at various time points from D-16, D-8, H + 1, and CT animals (right panel). Samples reaching 50% or more neutralization were considered to be seroneutralizing. The pink square at day 12 in the D-16 graph indicates the percentage of seroneutralization with a 1/200 dilution. Source data are provided as a Source Data file.

line). We also measured the measles (MeV)-specific IgG titers in all animals after challenge. The CT animals had measurable MeV-specific IgG titers at the time of challenge and during the rest of the experiment (Fig. 4E, gray line) but the D-16 and D-8 animals produced higher titers of MeV-specific IgG on day 0, especially the D-16 animals (Fig. 4E, red line), supporting a boost in anti-MeV immunity offered by the MeV-NP immunization. This boost was further supported by the increase in MeV-specific IgG observed between day 0 and day 12 in H + 1 animals (Fig. 4E, blue line). We also determined the MeV neutralization titers in animals on the day of challenge (Fig. 4F). While at day 0 H + 1 and CT animals had low or undetectable MeV-neutralizing antibodies, D-16 and D-8 animals had high titers of MeV-neutralizing antibodies.

We assessed the neutralization activity of the plasma collected at various times after the challenge. We fixed the limit of positive neutralization above 50% neutralization of LASV Josiah on Vero E6 cells (Fig. 4G). On the day of challenge, only animals D-16_3 and D-8_1 produced neutralizing antibodies and both continued to produce neutralizing antibodies until day 28. In the D-16 group, D-16_1 produced neutralizing antibodies on day 6 that persisted until the end of the experiment. The plasma of animal D-16_2 showed low neutralizing activity and only neutralized more than 50% of a LASV inoculum on days 12 and 28. In the D-8 group, D-8_3 produced neutralizing

antibodies from day 3 to day 28 and the plasma of animal D-8_2 only showed neutralizing activity on days 12 and 21. It is important to note that the plasma of all animals neutralized LASV at a relatively low dilution, 1/40, except for the plasma of D-16_1, which also neutralized 63% of LASV Josiah at 1/200 at day 12 (Fig. 4G, pink square). No animals in the CT and H + 1 groups produced antibodies with neutralizing activity.

## Short-term immunization with MeV-NP is sufficient to induce LASV-specific T-cell responses

We analyzed the T-cell responses induced after challenge by performing T-cell activation assays using overlapping peptides covering the full sequence of LASV GP and NP (Fig. 5 and Supplementary Fig. 3). CD4+ T cells producing IFN-γ in response to GP stimulation were detected as early as day 6 in the D-16 and D-8 animals and peaked on day 9 in both groups (Fig. 4A, left panel, red and orange bars). The percentage of GP-specific IFN-γ-producing CD4+ T cells then diminished until day 21 post challenge. CT animals showed a low percentage of GP-specific IFN-γ-producing CD4+ T cells on day 9 only (Fig. 5A, left panel, gray bars) and H + 1 animals showed a low percentage of GP-specific IFN-γ-producing CD4+ T cells between day 3 and day 12 (Fig. 5A, left panel, blue bars). Activated CD4+ T cells expressing CD154 in

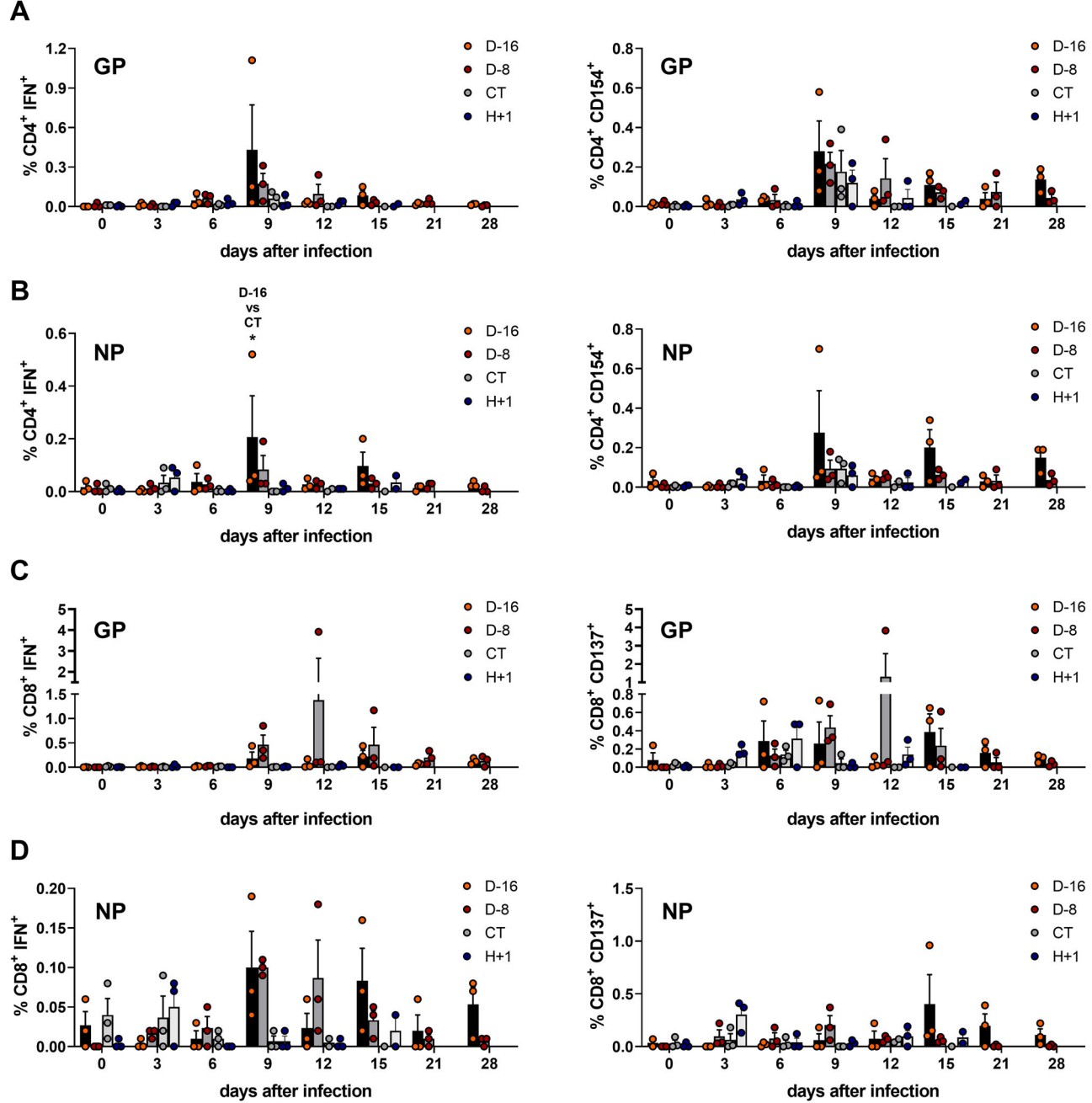

**Fig. 5 | LASV-specific T-cell responses induced after LASV challenge.**
**A** Quantification of the percentage of CD4+ T cells expressing IFNγ (left panel) or CD154 (right panel) in response to stimulation with LASV-derived GP peptides. **B** Quantification of the percentage of CD4+ T cells expressing IFNγ (left panel) or CD154 (right panel) in response to stimulation with LASV-derived NP peptides (D-16 vs CT, $p = 0.0382$). **C** Quantification of the percentage of CD8+ T cells expressing IFNγ (left panel) or CD137 (right panel) in response to stimulation with LASV-derived GP peptides. **D** Quantification of the percentage of CD8+ T cells expressing IFNγ (left panel) or CD137 (right panel) in response to stimulation with LASV-derived NP peptides. The percentage of T cells is presented according to the time after challenge after subtraction of the respective value measured for unstimulated T cells. Each bar represents the mean ± SEM of three samples. A Kruskal–Wallis test was used after testing for normality with a Shapiro–Wilk normality test. Statistical significance: *$p \leq 0.05$. Source data are provided as a Source Data file.

response to GP stimulation were detected from day 6 to day 28 in the D-16 and D-8 animals, with the percentage of CD154-producing CD4+ T cells peaking at day 9 (Fig. 5A, right panel, red and orange bars). In CT animals, CD154-producing CD4+ T cells in response to GP stimulation were only detected on day 9 (Fig. 5A, right panel, gray bars) while in H + 1 animals they were detected between day 3 and day 15 with a small increase at day 9 (Fig. 5A, right panel, blue bars).

We also detected, although at lower percentage, CD4+ T cells producing IFN-γ in response to NP stimulation in the D-16 and D-8 animals from day 0 to day 28 (Fig. 5B, left panel, red and orange bars).

The maximum percentage of NP-specific IFN-γ-producing CD4+ T cells was detected on day 9 in both D-16 and D-8 groups, with a higher percentage in D-16 animals than CT animals ($p < 0.05$). After day 9, the percentage of NP-specific specific IFN-γ-producing CD4+ T cells tended to decrease, except for a rebound at day 15 in the D-16 animals. A low percentage of CD4+ T cells producing IFN-γ in response to NP was only detected at days 0 and 3 in CT animals (Fig. 5B, left panel, gray bars) and at days 3, 9, 12 and 15 in H + 1 animals (Fig. 5B, left panel, blue bars). Activated CD4+ T cells expressing CD154 in response to NP stimulation were detected at various times between day 0 and day 28 in the D-16

and D-8 animals, with a peak on day 9 (Fig. 5B, right panel, red and orange bars) for both groups but a trend towards higher percentages in the D-16 animals. Low percentages of CD154-producing CD4+ T cells in response to NP stimulation were only detected at days 9 and 12 in CT animals (Fig. 5B, right panel, gray bars) and at days 3, 9, 12, and 15 in H + 1 animals (Fig. 5B, right panel, blue bars).

We measured a robust CD8+ T-cell response against GP by day 9 in the D-8 animals (Fig. 5C, left panel, orange bars) that peaked at day 12 then slowly decreased until day 28 (Fig. 5C, left panel, orange bars). The GP-specific CD8+ T-cell response was less intense in the D-16 animals (Fig. 5C, left panel, red bars) and we did not detect any GP-specific CD8+ T-cell response in the CT animals and H + 1 animals (Fig. 5C, left panel, gray and blue bars, respectively). We also measured an increasing percentage of GP-specific CD8+ T cells expressing CD137 in response to GP in the D-8 animals between days 6 and 12 (Fig. 5C, right panel, orange bars), mirroring the percentage of GP-specific IFN-γ positive CD8+ T cells. The percentage of CD137-expressing CD8+ T cells in response to GP also increased between days 6 and 21 in D-16 animals (Fig. 5C, right panel, red bars) and to a lesser extent between day 3 and day 12 in H + 1 animals (Fig. 5C, right panel, blue bars) but remained low in the CT animals (Fig. 5C, right panel, gray bars).

The CD8+ T-cell response against NP was very low, with only approximately 0.1% of T cells expressing IFN-γ on day 9 in both the D-16 and D-8 groups (Fig. 5D, left panel, red and orange bars), despite the higher percentage of CD8+ T cells expressing CD137 in response to NP on day 9 for the D-8 animals and day 15 for the D-16 animals (Fig. 5D, right panel, red and orange bars).

## Phenotypes of T cells and NK cells during the course of LASV infection

We monitored the evolution of the number of the various cell types within leukocytes and analyzed the phenotypes of circulating CD8+ and CD4+ T cells, as well as those of NK cells, to gain further insight into the cellular immune responses induced after the challenge. As previously reported[23–25], early and transient lymphopenia affecting all cell subpopulations was observed in all animals from day 3 after infection (Supplementary Fig. 4). In the CT and H + 1 animals, lymphocyte and NK cell counts remained low during the course of the disease, whereas the number of monocytes, in particular granulocytes, was restored between days 6 and 9. In contrast, the number of CD4+ T cells, B cells, monocytes, and granulocytes rapidly returned to basal levels, except for a transient drop by day 21, in all immunized animals, with no major differences observed between the D-16 and D-8 animals. There was a large increase in the number of CD8+ T cells and NK cells between days 9 and 15 in the D-16 and, in particular, D-8, animals, indicating a substantial enrichment of cytotoxic cells in the immunized animals.

We observed a transient increase in the percentage of CD8+ and CD4+ T cells (Fig. 6A, B) and in that of NK cells in the H + 1 animals and to a lesser extent in CT animals (Fig. 6C) on day 6. Similarly, the percentage of cells expressing CD134 transiently increased on day 9 among CD8+ T cells in the CT and H + 1 animals (Fig. 6A) and among CD4+ T cells in the CT, H + 1 and D-16 animals (Fig. 6B). The proportion of CD8+ and CD4+ T cells expressing CD279 (PD1) increased from day 9 in the H + 1, CT and D-16 animals and, to a lesser extent, in the D-8 animals. More than one-third of the CD8+ T cells expressed NKp80 between days 9 and 15 in the D-8 animals, whereas this population was only slightly enriched in the D-16 animals and almost completely disappeared during the course of infection in the CT and H + 1 animals (Fig. 6A). A large proportion of CD8+ T cells expressed the proliferation marker KI67 on days 9 and 12 in D-16 and D-8 animals, whereas such proliferation was only observed in the one surviving CT animal on day 14 and not in the H + 1 animals. By contrast, there was no increase in the percentage of CD4+ T cells expressing KI67 (Fig. 6B). All animals showed a transient increase in KI67 expression by NK cells, with approximately half proliferating by day 9 (Fig. 6C). The proportion of

CD8+ T cells expressing granzyme B (GrzB) was elevated in all animals between days 3 and 9, as well as day 28 in the D-16 animals (Fig. 6A). The expression of perforin was also highly elevated in all animals on day 9, with the highest percentage measured in the D-8 animals and the lowest in the H + 1 animals. The expression of GrzB by NK cells almost disappeared by day 12 in the immunized animals, whereas the percentage of cells expressing perforin increased in all animals during the course of infection, with a peak on day 9 (Fig. 6C).

We also analyzed the evolution of memory phenotypes during the experiment (Fig. 7, Supplementary Figs. 5, 6). The number of all subtypes of memory T cells dropped dramatically in all animals by day 3, indicating that the lymphopenia observed at this time (see below) affected all memory subtypes. Among CD8+ T cells from the CT and H + 1 animals, the number of the various subtypes remained low during the course of the disease, except for naive/pre-effector (pE)1 (CD45RA+ CD28+ CD27+) and central memory (CM)/effector memory (EM)1 (RA−28+27+) CD8+ T cells for the CT animals only, which were partially restored (Fig. 7 and Supplementary Fig. 6). By contrast, in immunized animals, the number of naive/pE1, CM/EM1, EM2 (RA−28−27+), pE2 (RA+28−27+), EM4 (RA−28−27−), and terminally-differentiated effector memory (EMRA) (RA+28−27−) cells increased dramatically between days 9 and 15, whereas EM3 (RA−28−27−) and RA+28+27− CD8+ T cells were transiently enriched on day 12 in D-8 animals (Supplementary Fig. 6). In terms of the proportion of memory subtypes within CD8+ T cells, the D-16 animals showed an enrichment in EMRA cells on days 3 to 6 and CM/EM1 cells on day 9, followed by a large increase in the proportion of EM2 and EM3 cells from day 9 (Fig. 7). The D-8 animals showed a similar change in the proportion of EM2 CD8+ T cells, together with an even larger increase in EM3 cells that represented half of all CD8+ T cells by day 12. These animals also showed an increase in the percentage of RA+28+27− CD8+ T cells on days 3 and 28. Among CD4+ T cells, only the EM4 subtype was enriched in all animals from days 3 to 6 DPI. In the CT and H + 1 animals, only the number of naive/pE1− and RA+28+27− CD4+ T cells increased after day 3. By contrast, the number of all memory CD4+ T-cell subtypes increased in the immunized animals from day 6, but more rapidly in the D-8 animals for EM2 and EM3 cells (Supplementary Fig. 6).

## Short-term vaccination shapes immune responses at the transcriptional level

Based on our results, both humoral and T-cell responses may have participated in protection in immunized monkeys. We tried to better understand the mechanisms of protection in D-16 and D-8 animals by comparing the transcriptomic profiles of their PBMCs. Given the absence of benefit of the post exposure vaccination, samples from the group H + 1 were not included in the analyses but we included samples of the CT group. We extracted cellular RNA from PBMC samples collected at various times post-infection and performed RNA sequencing (RNA-seq). RNA-seq data were then analyzed to generate heat maps of the most differentially expressed genes for particular responses (Fig. 8). Genes associated with the antiviral response were upregulated on days 3 and 6 post challenge in all animals relative to day 0 ($p < 0.0001$) (Fig. 8A). However, such upregulation was more intense in the CT than D-16 and D-8 animals ($p < 0.0001$). By day 9, the expression of antiviral genes had decreased in the D-16 and D-8 animals, whereas it remained elevated in the CT animals at days 9 and 12 relative to day 0 ($p < 0.0001$).

The expression of many cytokines and chemokines was also upregulated following challenge (Fig. 8B). In the three groups, a subset of genes, including *CCL8*, *TNFSF13B*, *LTβR*, *CSF1*, *TNFRSF1A*, *CCL2*, *CSF2RB*, *IL27*, *IL1RN*, and *IL15RA*, was upregulated on days 3 and 6, particularly in the CT animals ($p < 0.0001$ at day 3, $p < 0.001$ at day 6). After day 6, these genes were no longer upregulated in the D-16 and were downregulated in the D-8 animals, but remained upregulated

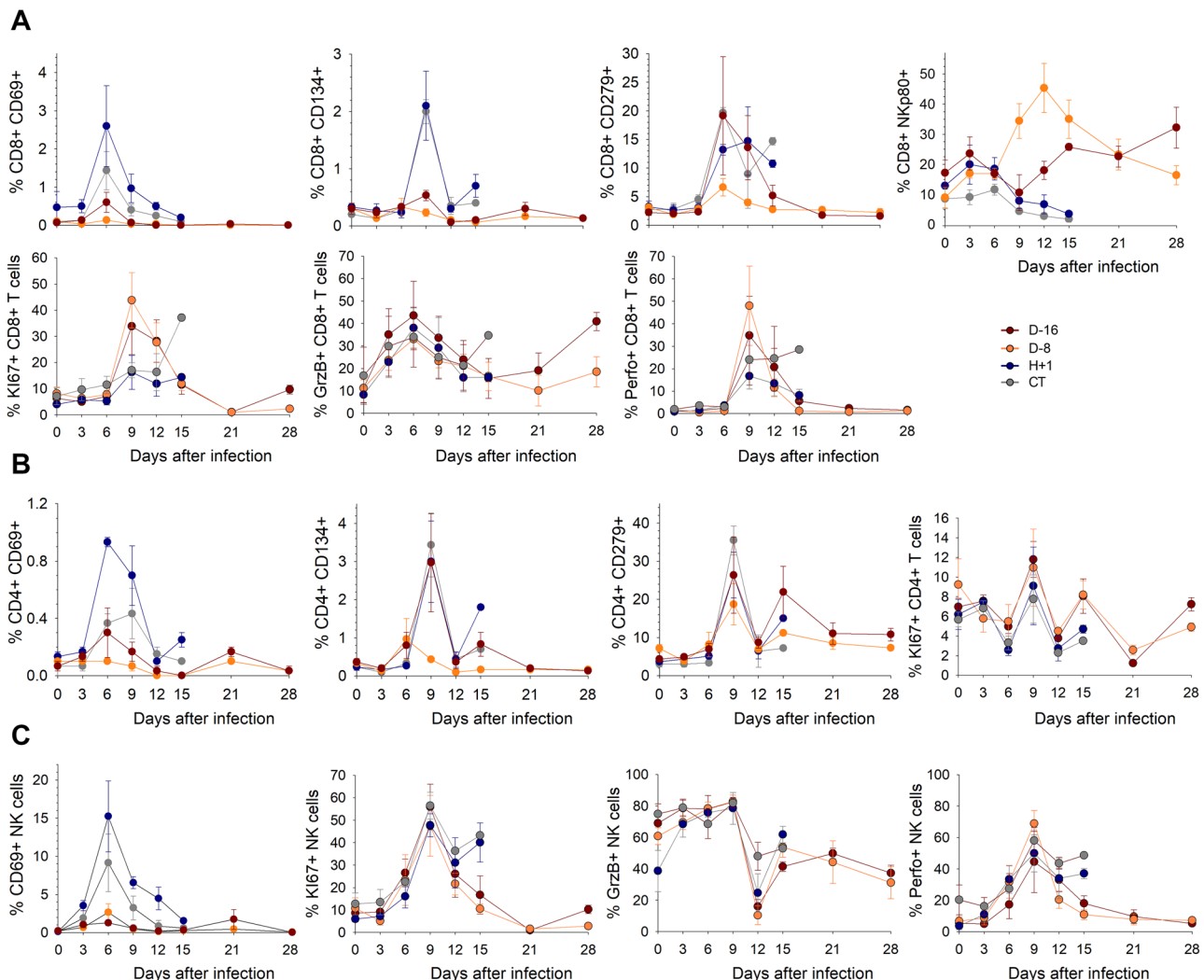

**Fig. 6 | Evolution of T-cell and NK cell populations during the course of LASV infection. A** Percentage of CD8$^+$ T cells expressing CD69, CD134, CD279, NKp80, KI67, granzyme B (GrzB), or perforin (Perfo) in the blood samples of animals according to the time after the LASV infection measured by flow cytometry. Data are presented as mean values ± SEM of $n = 3$ independent samples. **B** Percentage of CD4$^+$ T cells expressing CD69, CD134, CD279, or KI67 in the blood samples of animals according to the time after challenge measured by flow cytometry. Data are presented as mean values ± SEM of $n = 3$ independent samples. **C** Percentage of NK cells expressing CD69, KI67, granzyme B (GrzB), or perforin (Perfo) in the blood samples of animals according to the time after challenge measured by flow cytometry. Data are presented as mean values ± SEM of $n = 3$ independent samples.

in the CT animals until day 12. Starting at day 9, another subset of cytokine/chemokine genes was upregulated in the CT animals. This subset included the pro-inflammatory genes *IL2RA, IL21, IFNγ, IL18, IL18R1, IL18RAP, CXCR2, CXCR1, IL1R1, IL1R2, CSF3R, IL1RAP, IL6, OSMR,* and *IL4R*.

Concomitant to the induction of cytokine/chemokine gene expression, the expression of genes associated with the monocyte response was also affected by the challenge (Fig. 8C). In D-16 animals, these genes were transiently upregulated on day 3 relative to that on day 0 ($p = 0.00362$). In D-8 animals, the expression of these genes was upregulated by the infection on days 3 ($p = 0.00077$) and 6 ($p = 0.0004$) and then considerably downregulated by day 9 relative to that on day 0 ($p < 0.0001$) and of the other groups ($p < 0.0001$). In CT animals, genes associated with the monocyte response were strongly upregulated on day 3 ($p < 0.0001$) and day 6 ($p < 0.0001$) relative to that on day 0 and of the other groups ($p < 0.0001$). After day 6, most monocyte-related genes were downregulated in the CT animals, except for a subset of genes that were upregulated until day 12, including the genes *FBP1, ALOX5,* and *MGST1,* known to be involved in lipid metabolism and the production of leukotrienes.

There were strong differences between groups in the upregulation of genes associated with the B-cell response (Fig. 9A). In D-16 animals, the B-cell response was not upregulated at any time after challenge and was strongly downregulated after day 6 ($p < 0.0001$). In D-8 animals, most genes associated with this response were already upregulated on day 0 relative to the other groups ($p < 0.0001$) and remained highly expressed until day 6 ($p < 0.0001$). The expression of these genes decreased by days 9 and 12 before increasing again at day 15, suggesting a second phase of B-cell activation. In CT animals, genes associated with the B-cell response were mostly downregulated after day 0, except for a subset of genes including *ADAM28, PTPRK, STAP1,* and *IGHM,* which were upregulated from day 6 to day 12.

Striking differences between groups were also observed at the level of T-cell response-related mRNA levels (Fig. 9B). In CT animals, we observed a strong downregulation of T-cell associated genes at day 3 ($p = 0.00022$) and day 6 ($p < 0.0001$) compared to day 0. This downregulation was also observed in D-16 and D-8 animals but was less intense, more transitory. The D-16 and D-8 animals showed upregulation of T-cell associated genes starting from day 9, and the upregulation of these genes was particularly intense in D-8 animals at days 9 and

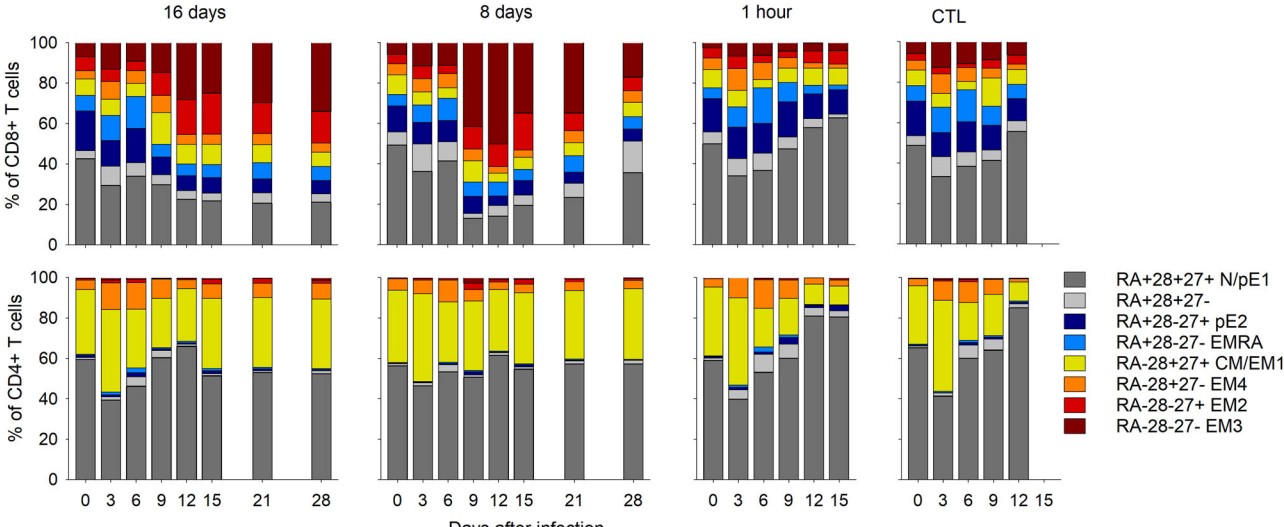

**Fig. 7 | Evolution of memory cell subpopulations during the course of LASV infection.** Proportions of memory cell subpopulations among CD8+ (upper panels) and CD4+ (lower panels) T cells in D-16, D-8, H+1, and CT animals according to the time after challenge measured by flow cytometry. Source data are provided as a Source Data file.

12 ($p < 0.0001$) relative to that on day 0 and that of the other groups ($p < 0.0001$ for D-8 vs D-16 and $p = 0.00086$ for D-8 vs CT at day 9, $p < 0.0001$ for D-8 vs D-16 and $p = 0.0017$ for D-8 vs CT at day 12).

We also used the transcriptomic data to carry out a deconvolution to infer the proportion of cell types in the PBMC samples over time (Supplementary Fig. 7 and Fig. 10). Using an LM22 matrix and CIBER-SORT methods[28], we inferred the proportion of 22 cell types in the PBMC samples (Supplementary Fig. 7). For the three groups, monocytes, B cells, and T cells were the most highly represented cell types at all time points but their proportions varied between groups over time (Supplementary Fig. 7). The proportion of monocytes in the D-16 animals at day 0 than in the D-8 and CT animals but increased in all groups between days 0 and 3 (Fig. 9, upper left panel). After day 3, the proportion of monocytes decreased in the D-16 and D-8 animals but was statistically higher on day 9 in the D-16 animals compared to the D-8 animals ($p < 0.05$). In the CT animals, the proportion of monocytes increased until day 9 and remained high relative to that in the D-8 animals. On day 0, the proportion of activated NK cells was low in all groups (Fig. 9, upper middle panel), but tended to increase rapidly in at least two D-16 animals by days 3 and 6 before decreasing from day 6 to day 15, even if the differences were not statistically significant. In the D-8 group, the proportion of activated NK cells was smaller between days 6 and 12 in the three animals. In the CT group, only one animal showed an increased activated NK cell signature at days 6, 9, and 12. There was no difference in the proportion of plasma cells between groups and their numbers did not markedly change over time (Fig. 9, upper right panel). However, the proportion of memory B cells was particularly high in the D-8 animals, especially on day 6, when the proportion of these cells was significantly higher ($p < 0.05$) than in the CT animals (Fig. 9, center left panel). Although the proportion of memory B cells remained high in the D-16 and D-8 animals after day 6, it decreased significantly after day 6 in the CT animals relative to day 0 ($p < 0.01$). For CD8 T cells, the proportion tended to increase after day 6 in the D-16 and D-8 animals and remained high after that time point (Fig. 9, center middle panel). At day 9, the D-8 animals had a significantly higher proportion of CD8 T cells than at day 0 ($p < 0.05$). The proportion of CD8 T cells did not evolve over time in the CT animals. However, the proportion of naive CD4 T cells increased considerably over time after challenge (Fig. 9, center right panel) and was significantly higher than in the D-16 and D-8 animals on day 12 ($p < 0.05$), for which the proportion of these cells remained low. We did not

observe major differences in the proportions of activated memory CD4 T cells (Fig. 9, lower left panel) or follicular helper T cells (Fig. 9, lower middle panel) between groups at the various time points, although the D-8 animals tended to have higher proportions of these cells at day 9. The D-16 and D-8 animals had significantly lower proportions of regulatory T cells at day 0 and day 3 than the CT animals (D-16 vs CT, $p < 0.05$; D-8 vs CT, $p < 0.01$) (Fig. 9, lower right panel). Then, the proportions of regulatory T cells were maximal on day 9 for each group and significantly different from that on day 0 on days 9 and 12 for the D-8 animals.

## Discussion

Vaccination against LASV is a major challenge in Western Africa, where yearly epidemics remain uncontrolled. We previously demonstrated the efficacy of MeV-NP against Lassa fever in cynomolgus monkeys immunized one month or one year before a LASV infection[23,24], as well as its protective efficacy against recently isolated and highly divergent Lassa strains from lineages II and VII[24,27]. Here, we demonstrate that MeV-NP can protect cynomolgus monkeys with similar or even higher efficacy when the monkeys are immunized 16 or 8 days before a LASV infection. This is particularly interesting, as most LASV outbreaks are limited to geographical clusters[29,30], for which infected rodents are the main source of viral dissemination in the human population[7,31]. In this context, a vaccine capable of rapid protection of an exposed population in an outbreak setting would be highly beneficial. Given the estimated 10-day incubation period of Lassa fever, generalizing vaccination at the beginning of LASV outbreaks in delimited geographical zones could considerably reduce the burden of Lassa fever.

The protection offered to macaques by MeV-NP in a short-term vaccination setup is particularly efficient. We could not detect any infectious virus in one D-16 and two D-8 animals and only trace amounts of viral RNA in some of their organs. Moreover, animals D-16_2 and D-8_2 showed no fever and no alteration of biochemical parameters. The other MeV-NP vaccinated animals showed some clinical signs but were also strikingly protected. Interestingly, we observed notable differences between the D-16 and D-8 animals. The D-8 animals had a less prolonged fever, a more transitory increase in CRP levels, and lower viral loads than the D-16 animals, even if these differences were not statistically significant. In previous studies, we demonstrated that MeV-NP could protects cynomolgus monkeys against Lassa fever after a single shot given a year or a month before

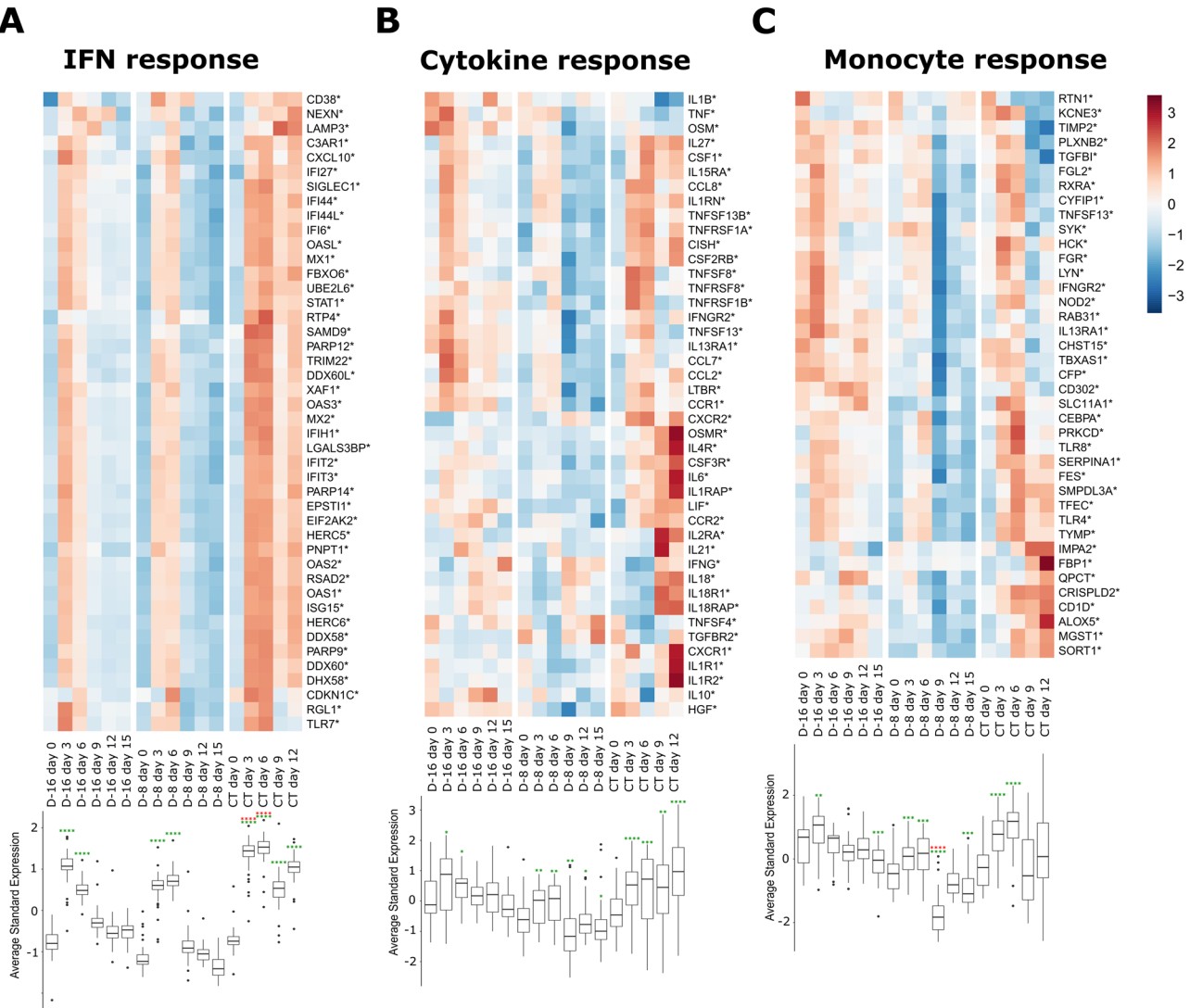

**Fig. 8 | Evolution of the innate transcriptomic signatures in PBMC samples after LASV challenge over time.** Gene expression heatmap of three gene sets: **A** antiviral response, **B** cytokine response, and **C** monocyte response. Each time point represents the mean of $n = 3$ independent samples. The color of the cell represents the average standardized (centered and scaled) gene expression and the intensity of standardized expression is given by the scale. Bottom: Boxplot representation of the standard average expression (Standard aver. expr.) of the genes for each time point in the three gene sets (central line, median; hinges, first and third quartiles; whiskers, largest or smallest value no further than 1.5 times the inter-quartile range from the hinge). Outlying data are plotted individually. Mean values for each time point were statistically compared to assess their difference using the non-parametric two-samples unpaired Wilcoxon test with the Benjamini–Hochberg correction for multiple testing. The red asterisks represent significant differences between groups for the same time point and the green asterisks represent significant differences between time points for the same group. Statistical significance: *$p \leq 0.05$, **$p \leq 0.01$, ***$p \leq 0.001$, and ****$p \leq 0.0001$.

the LASV challenge[23,24]. With all regimen, we observed a very efficient control of the virus replication with no detectable viral titers in the blood and no clinical signs of severe Lassa fever. Therefore, MeV-NP offers a large window of protection against Lassa fever, from 8 days up to a year after a single immunization. Importantly, MeV-NP conferred comparable protection in animals that were naive or pre-immune to MeV, supporting no effect of the MeV pre-existing immunity on the efficacy of MeV-NP, at least in the D-16 and D-8 regimen. In MeV pre-immune animals, the MeV-NP vaccination rather boosted the humoral response against MeV, as observed in humans vaccinated with the MV-CHIK vaccine[32]. More recently, pre-existing immunity was proposed to dampen the immunogenicity of V591, a measles-based SARS-CoV2 vaccine[33]. Although our data suggest that MeV-NP may not be affected by pre-existing immunity, this aspect needs to be studied in detail in humans. MeV-NP has been tested in a phase I clinical trial under the name MV-LASV (NCT04055454) and the results will bring further insights on pre-existing MeV immunity.

The apparently better efficacy of the D-8 over D-16 vaccination could be attributed to a strong B-cell response. Indeed, genes associated with the B-cell response were particularly upregulated in the D-8 animals at day 0 relative to the other animals. Nevertheless, the precocity of this response likely represents the B-cell response against MeV antigens. Indeed, MeV-NP immunization offers a boost in the humoral response against MeV and the D-8 animals were still producing anti-MeV antibodies at the time of challenge. However, we observed a second phase of an elevated B-cell response in the D-8 animals between days 9 and 12, which was more modest in the D-16 animals. This second phase may represent the specific B-cell response against LASV antigens, which also coincided with the peak of LASV-specific IgG production in both groups. Despite the production of LASV-specific IgG, low titers of neutralizing antibodies were detected in MeV-NP-immunized animals and the best controllers showed the lowest neutralizing activity. This observation supports the minimal benefit of developing neutralizing antibodies in vaccine-induced

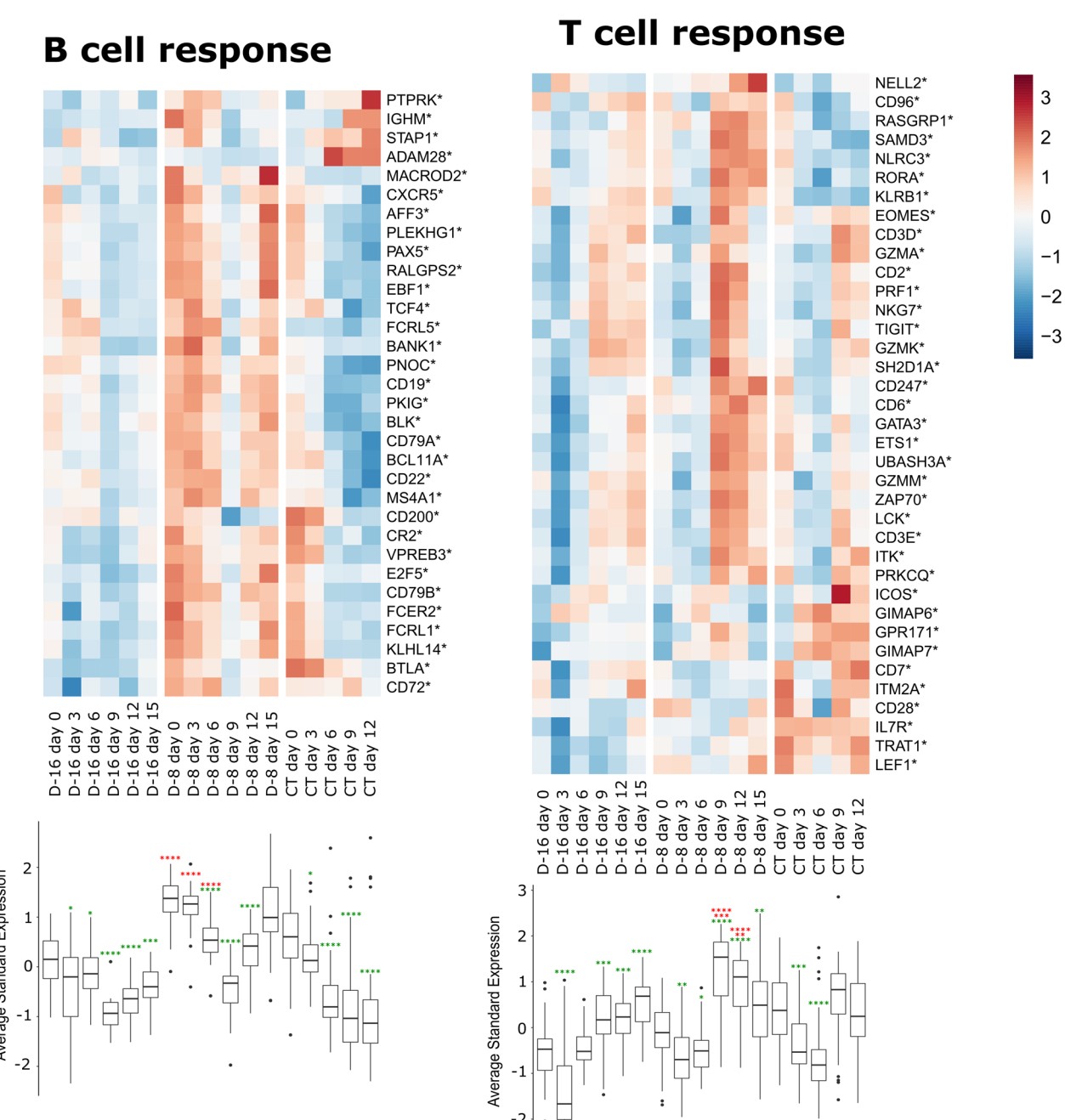

**Fig. 9 | Evolution of the adaptive transcriptomic signatures in PBMC samples after LASV challenge over time.** Gene expression heatmap of two gene sets: **A** B-cell response and **B** T-cell response. Each time point represents the mean $n = 3$ independent samples. The color of the cell represents the average standardized (centered and scaled) gene expression and the intensity of standardized expression is given by the scale. Bottom: Boxplot representation of the standard average expression (Standard aver. expr.) of the genes for each time point in the two gene sets (central line, median; limits, first and third quartiles; whiskers, largest or smallest value no further than 1.5 times the interquartile range from the hinge). Outlying data are plotted individually. Mean values for each time point were statistically compared to assess their difference using the non-parametric two-samples unpaired Wilcoxon test with the Benjamini-Hochberg correction for multiple testing. The red asterisks represent significant differences between groups for the same time point and the green asterisks represent significant differences between time points for the same group. Statistical significance: $*p \leq 0.05$, $**p \leq 0.01$, $***p \leq 0.001$, and $****p \leq 0.0001$.

protection for Lassa fever, as suggested in many studies[16,23,24,34,35]. In contrast, non-neutralizing antibodies may play a role in protection as maximal IgG titers were reached by the time of virus clearance, as observed when animals were immunized by MeV-NP one month or one year before the challenge[23,24]. While we were not able to link the protective effect of non-neutralizing antibodies to antibody-dependent cell cytotoxicity (ADCC) in our previous experiments[24], ADCC has been shown to be important for the protection conferred by a recombinant Lassa-Rabies vaccine in mice[34].

We observed a massive increase in the number of circulating CD8+ T cells in surviving animals from day 9 after infection and an enrichment of this cell subtype was also detected in these animals by

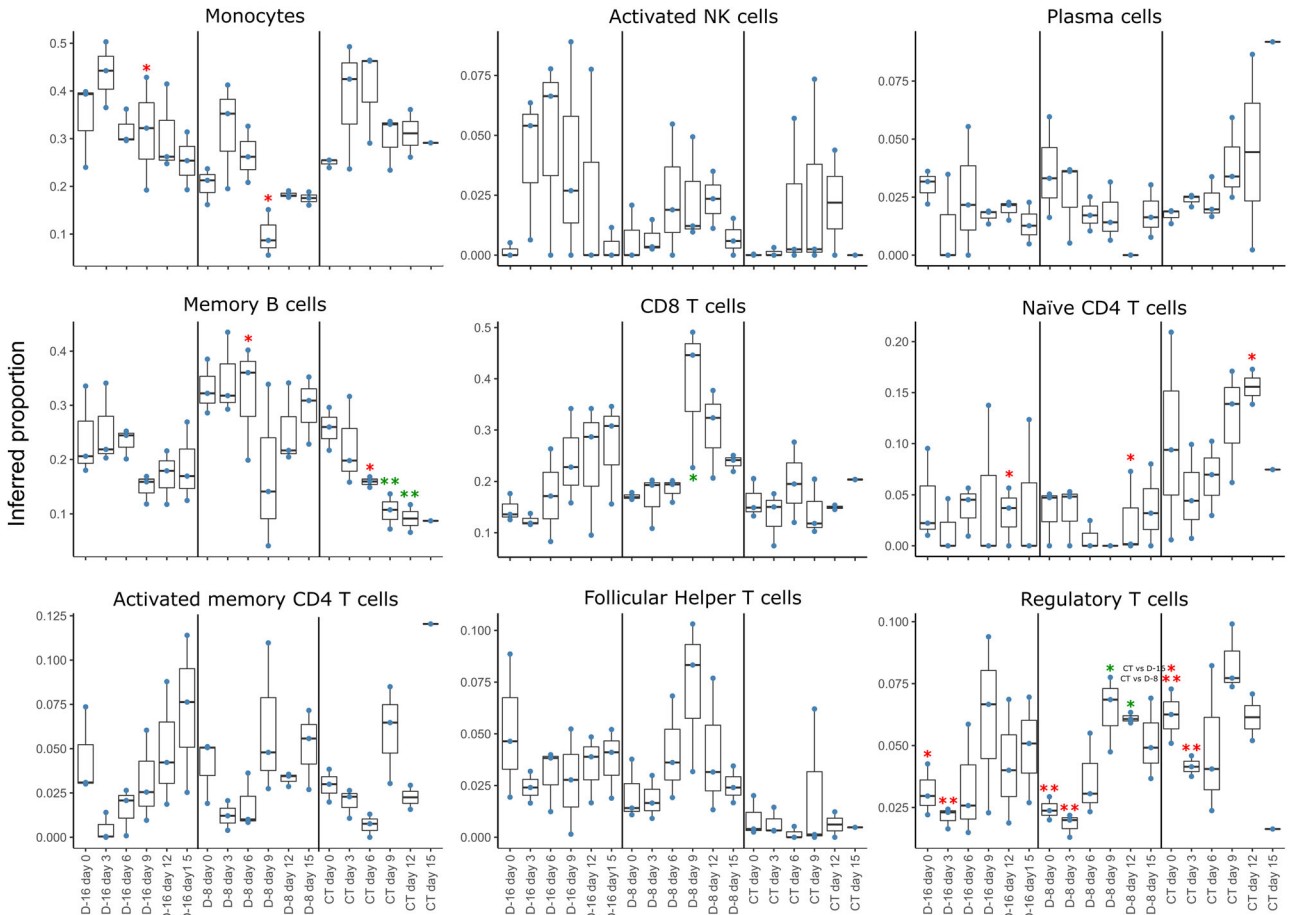

**Fig. 10 | Evolution of the cell-type composition estimates in PBMC samples over time based on transcriptome data.** Inferred proportion of monocytes, activated NK cells, plasma cells, memory B cells, CD8[+] T cells, naive CD4[+] T cells, activated memory CD4[+] T cells, follicular helper CD4[+] T cells, and regulatory T cells in PBMC samples collected at various time points after the LASV infection (central line, median; limits, first and third quartiles; whiskers, largest or smallest value no further than 1.5 times the interquartile range from the hinge). Blue dots represent individual proportion estimates ($n = 3$). A two-sided ANOVA was performed, with Tukey multi-comparison tests, to compare the inferred proportions between time points. The red asterisks represent significant differences between groups for the same time point and the green asterisks significant differences between time points for the same group. Statistical significance: *$p \leq 0.05$, **$p \leq 0.01$, ***$p \leq 0.001$, and ****$p \leq 0.0001$.

the deconvolution analysis. This increase was more intense in D-8 animals. At this time point, a large proportion of CD8[+] T cells were proliferating and expressed perforin, GrzB and, in the D-8 animals, the activation marker NKp80. Overall, these results suggest that a robust cytotoxic T-cell response is induced in protected but not in unprotected animals. LASV-specific CD8[+] T cells were induced at this time point, as confirmed by the production of IFNγ and the expression of CD137 in response to LASV-derived peptides. Such a specific CD8[+] T-cell response was already observed during the previous experiments with the MeV-NP vaccine[23,24], suggesting that the kinetics of T-cell activation are not modified by short-term immunization. The transcriptomic profiles measured in PBMCs by gene-set and deconvolution analyses showed the activation of CD8[+] T cells to be much more intense in the D-8 than D-16 animals. This robust CD8[+] T-cell response, which correlates with the very efficient protection observed in this group, suggests that CD8[+] T cells probably play a major role in the vaccine-induced control of LASV. CD4[+] T cells also appeared to be activated in surviving animals. The number of circulating CD4[+] T cells was restored to basal levels from day 9 after infection in these animals but no significant cell proliferation occurred. An enrichment in the proportion of follicular helper T cells was suggested in the D-8 animals, whereas the proportion of regulatory T cells significantly decreased in the first week after challenge in the surviving animals. LASV-specific CD4[+] T cells were circulating

in immunized animals from day 9 after challenge, as demonstrated by the production of IFNγ and the expression of CD154 in response to LASV-derived peptides. CD4[+] T-cell activation was also measured at day 9 in the unprotected animals, as suggested by the expression of CD154, CD134, and CD279, as well as by the transcriptomic data. The transcriptomic data indicate enrichment of naive T cells, regulatory T cells, and activated memory T cells. The presence of LASV-specific CD4[+] T cells in the CT and H + 1 animals is consistent with an induced humoral response. The CD4[+] response in the D-16 animals against NP was more intense at day 9 than that in the D-8 animals. Although the NP-specific CD4[+] T-cell response may contribute to protection, its benefit appears to be less than that of the CD8[+] T-cell response against GP, as D-16 controlled LASV replication less efficiently. The better protective effect of CD8 responses over CD4 responses also corroborates our previous observations[23]. Moreover, no substantial change was observed for CD4[+] T cells in the phenotype of memory cells, whereas memory CD8[+] T cells were enriched in surviving animals, with a massive increase in the number of circulating CM/EM1, EM2, EM3, EM4, and EMRA CD8[+] T cells. These results reinforce the hypothesis that CD8[+] T cells are crucial for the control of LASV infection. In CT and H + 1 animals, the slight changes in the proportion of CD8[+] T-cell subtypes did not translate into an increase in the number of these cells among PBMCs. Adoptive transfers or T-cell depletion experiments would allow to clearly demonstrate the role of

different T-cell subsets in protection. However, these experiments were out of the scope of this study.

MeV-NP failed at protecting cynomolgus monkeys in a therapeutic setup. All H + 1 animals developed a similar disease and mounted similar anti-LASV immune responses than control animals that failed to control the LASV infection. However, H + 1 animals responded to the vaccine as demonstrated by the boost in the production of anti-MeV IgG after immunization. This observation suggests that innate or adaptive immune responses against the MeV vector do not help controlling LASV replication, at least in a therapeutic setup. This contrasts with what was observed with VSV platform in the context of filovirus infections. Indeed, monkeys vaccinated with a recombinant VSV expressing the Ebola (EBOV) glycoprotein, VSV-EBOV, at 1 h and 24 h post an EBOV challenge were protected at 66% but controls vaccinated with VSV-MARV were also protected at 66% against EBOV, suggesting that innate responses to the VSV backbone may participate in protection[36]. In separate studies, a recombinant VSV expressing the Marburg (MARV) glycoprotein, VSV-MARV, protected macaques when administered as short as 20–30 min after a MARV challenge[37,38] and protection was partially associated with early innate responses based on transcriptomic analyses[38]. We had previously demonstrated that MeV-NP immunization induces an early innate response during the first 7 days following vaccination that completely disappears after day 7[23]. Clearly, this response was not sufficient to help controlling the LASV infection and was most likely overwhelmed by the innate response against the LASV infection in H + 1 animals[25].

The total absence of protection may be explained by the timing of the immune response against the LASV antigens. Indeed, the LASV-specific immune responses observed after an MeV-NP immunization in monkeys were mild in our previous experiments, with inconsistent detection of IgM and IgG before day 14 and weak T-cell responses also detected after day 14[23,24]. By this time, LASV-infected monkeys are generally in the terminal phase of the disease, which may explain the absence of protection. So far, only two vaccines, the Mopeia–LASV reassortant ML29 and the LASV replicon particles VRP, were able to protect rodents from death after a therapeutic vaccination, but the animals still developed clinical manifestations of the disease and no correlate of protection was identified[39,40]. More recently, Cross and colleagues tested the short-term efficacy of rVSVΔG-LASV-GPC against a heterologous LASV strain in cynomolgus monkeys vaccinated 7 or 3 days before challenge[18]. In both cases, all rVSVΔG-LASV-GPC-immunized monkeys survived while rVSVΔG-EBOV-GP-immunized monkeys died within two weeks. Transcriptomic analyses on PBMCs revealed a stronger unspecific response induced by the −3 days immunization than by the -7 days immunization but this unspecific response did not translate to better control of virus replication as viremia was controlled earlier in -7 days animals than in -3 animals. Protection was rather associated with specific humoral and cellular responses against LASV. We did not evaluate the short-term efficacy of MeV-NP against heterologous lineages. Considering that vaccinating monkeys one month before a heterologous challenge was protective[24] and that short-term vaccination seemed at least as efficient as a one-month vaccination, we believe that MeV-NP would protect efficiently against LASV strains from other lineages.

A major limitation of this study is the absence of control animals vaccinated with an irrelevant MeV vaccine. We cannot host >12 animals in our BLS-4 animal facility, limiting the number of control animals that can be included. In absence of a clear incentive to include an unspecific MeV control group receiving vaccine at D-16, D-8, or H + 1, we decided to keep the control group unvaccinated. The data presented here clearly reiterate the benefit of a MeV-NP vaccination even when given shortly before challenge. While we cannot formally rule out that unspecific responses to the vector may have participated in protection, we believe this aspect to be negligible for two reasons. First, we previously demonstrated that unspecific immune responses against the vaccine are only detectable for a week in MeV-NP immunized animals[23]. Second, unspecific immune responses against MeV-NP had no effect on LASV replication in H + 1 animals. As all animals were pre-immune to MeV, the MeV-NP immunization boosted the adaptive response against MeV in D-16, D-8, and H + 1 animals, highlighted by the kinetics of MeV-specific IgG. T-cell responses against MeV are also likely to be induced by the MeV-NP immunization, but it appears unlikely that they would play a major role in the control of LASV replication. The memory CD8 and CD4 responses observed after challenge in D-16 and D-8 animals also support the specificity of the immune response. Indeed, these memory responses should be delayed by 8 days when comparing these two groups if they were MeV-specific. But the memory T-cell responses observed after the LASV challenge have similar kinetics, indicating their specificity towards LASV.

As observed in clinical trials with the MV-CHIK vaccine[32], MeV-NP boosted the immune response against MeV, and particularly the production of MeV-neutralizing antibodies. Being a bivalent vaccine would be an asset for MeV-NP. Indeed, West African countries have recently faced issues in their measles vaccination campaign due to the Ebola and COVID-19 outbreaks[41,42]. MeV-NP could potentially offer protection against two viral diseases, providing immunity against LASV and boosting or initiating immunity against measles. MeV-NP is as attenuated as the MeV Schwarz vaccine strain in monkeys[23] and other constructs based on the same vector were found to be safe and well tolerated in multiple phase 1 and 2 clinical trials[32,33,43]. Regarding individuals with different degrees of moderate immune suppression, an endemic phase Ib or phase II trial should help determine the safety of the MeV-NP vaccine. Importantly, MeV vaccination is recommended for HIV patients without severe immunosuppression[44] and the MeV vaccine is safe in HIV-infected children[45].

In this study, we have demonstrated that a single shot of MeV-NP can protect against Lassa fever when administered 16 or even 8 days before a LASV infection. This short time to protection offers the possibility to vaccinate people at risk during an outbreak and to reduce the burden of Lassa fever. The strong protection offered by MeV-NP eight days before an infection also suggests that a shorter time between vaccination and challenge may be possible.

## Methods
### Study design
This study aimed to determine the efficacy of a measles-based Lassa fever vaccine, MeV-NP, administered at reduced times before Lassa virus exposure in cynomolgus monkeys. The animals were all 2.5- to 3-year-old male cynomolgus monkeys (*Macaca fascicularis*) from Mauritius Island weighing 3.5–4 kg at the time of vaccination. We immunized intramuscularly (IM) two groups of three monkeys with $2 \times 10^6$ TCID$_{50}$ of MeV-NP at 16 (D-16) or 8 (D-8) days before a lethal challenge with the prototypic Lassa strain Josiah. A third group of three monkeys did not receive MeV-NP and served as a control group. Notably, all monkeys had received two shots of a measles live attenuated vaccine (Serum Institute of India) at least one year before entering the study and were therefore considered to be pre-immune to measles virus. Approximately 1 month prior to vaccination, monkeys were implanted with an abdominal transponder to provide a live recording of the body temperature. Surgeries were performed in SILABE (Strasbourg) as well as the immunization of the D-16 group. Ten days prior to challenge, monkeys were transported to the BSL-4 Inserm P4-Jean Mérieux laboratory (Lyon), where each group was housed separately from the others. After two days in the BSL-4 laboratory, the D-8 animals were immunized with MeV-NP. The LASV challenge was performed by subcutaneous (SC) injection of 1,500 FFU of LASV strain Josiah. Prior to challenge, blood was drawn and oral and nasal swabs were collected from anesthetized animals. Anesthesia was also performed every three days for the first 15 days and then once a week until the end of the experiment to collect the same sample types and

perform a medical exam and determine the clinical scores. Clinical scores were attributed based on the body temperature, weight, dehydration, bleeding, petechiae, stool aspect, and reactivity. A clinical score of 15 or above required killing of the animals to limit suffering. Other limit points also required euthanasia, such as a body temperature <35.8 °C for a vigil animal, a post anesthesia coma lasting more than two-and-a-half hours, or convulsions, with or without balance issues. All procedures were approved by the Comité Régional d'Ethique en Matière d'Expérimentation Animale de Strasbourg (2018100414445313) and the Comité d'Ethique pour l'Expérimentation Animale CELYNE (2020061215142330).

### Cell cultures, virus, and infections
Vero NK and Vero E6 (ATCC CRL-1586) cells were maintained in Glutamax Dulbecco Modified Eagle's Medium (DMEM, Life Technologies) supplemented with 5% fetal bovine serum (FBS) and 0.5% penicillin-streptomycin. MeV-NP stocks were prepared as described elsewhere[23,24]. Briefly, Vero NK cells were infected at a MOI of 0.01 and viruses were harvested two to three days later by scraping cells into Opti-MEM I reduced-serum medium and two cycles of freeze-thawing. Titers were determined on Vero NK cells by $TCID_{50}$ titration. Vaccine stocks at the appropriate concentration were then prepared in 1% DMEM. The stock of LASV strain Josiah (a gift of S. Becker, Philipps-Universität, Marburg) was produced and titrated on Vero E6 cells. LASV Josiah was further diluted in phosphate-buffered saline (PBS) prior to challenge of the animals with 1,500 FFU by the SC route. Viral titers in plasma and organs were determined by titration on Vero E6 cells. Plaque reduction neutralization assays were also performed on Vero E6 cells as described elsewhere[23,24]. Plasma samples were diluted and mixed with 150 FFU of LASV for 1 h at 37 °C. Then the mixtures were added to Vero E6 cells for 1 h at 37 °C. Then carboxymethylcellulose diluted in DMEM was added and plates were incubated for seven days at 37 °C. After 7 days, the number of foci was calculated by focus-forming immunodetection using anti-LASV antibodies.

### Hematology and biochemistry
CD8 and CD4 T-cell, B-cell, NK cell, monocyte, and granulocyte counts were all determined by flow cytometry using antibodies directed against the following proteins: CD56 (560360, V540 mouse anti-human, clone B159, 1.25 μL), CD3 (560770, V500 mouse anti-human, clone SP34-2, 1.25 μL), CD45 (557803, FITC mouse anti-NHP, clone D058-1283, 5 μL), CD10 (557143, PE mouse anti-human, clone HI10α, 5 μL), CD20 (560735, PE-Cy™7 mouse anti-human, clone 2H7, 1.25 μL), CD4 (560836, Alexa Fluor[R] 700 mouse anti-human, clone L200, 1.25 μL), and CD8 (560179, APC-H7 mouse anti-human, clone SK1, 1.25 μL), all from BD biosciences, and an antibody from Miltenyi Biotec directed against NKp80 (130-094-845, APC mouse anti-human, clone 4A4.D10, 2.5 μL). After staining, cells were fixed and analyzed by flow cytometry using a 10-color Gallios cytometer (Beckman Coulter). Data were further analyzed using Kaluza software v2.1 (Beckman Coulter). Plasma concentrations of alanine aminotransferase (ALT), aspartate aminotransferase (AST), C-reactive protein (CRP), lactate dehydrogenase (LDH), albumin (ALB), and urea were measured using a Pentra C200 analyzer (Horiba Medicals).

### Quantitative RNA analysis
Viral RNA was extracted from fluids using the QIAamp Viral RNA mini kit (Qiagen) and from cells or tissues using the RNeasy mini kit (Qiagen). Quantitative RT-PCR was performed using the SensiFAST Probe No-ROX One-Step kit (Bioline) and NP-specific primers and probes for LASV Josiah (Forward: 5′-CTTTCACCAGGGGTGTCT-3′; Reverse: 5′-GTCACCTCAGACAATGGATGG-3′; Probe: 5′-TGAA-CATTCCAAGAGCC-3′). An NP-specific RNA standard produced in-house was used for quantification on a LightCycler 480 (Roche).

After a first step of RT at 45 °C for 10 min and an inactivation step at 95 °C for 2 min, 45 cycles of amplification (95 °C 1 min, 52 °C 30 s) were performed and the reaction cooled down to 37 °C. The limit of detection was 4285 copies per mL in plasma and swabs and 50 copies per mg in organs. Runs were analyzed with the LC480 software, 1.5.0 SP4 version (Roche).

### Transcriptomics
Total RNA was extracted from PBMCs collected on days 0, 3, 6, 9, and 12 using the RNeasy Mini Kit with DNase treatment (Qiagen). RNA samples were quantified using a Quantifluor (Promega) and qualified using the SS RNA system on a fragment analyzer (AATI). Poly(A)-capture and library preparations were performed as described elsewhere[23]. Sequencing was performed on a NextSeq 500 Flow Cell High Output SR75 instrument (Illumina) with nine samples per flow cell.

The evolution of the transcriptomic signatures in the PBMC samples after LASV challenge was studied using heatmaps for the user-defined list of genes generated with the *pheatmap* R package v 1.0.12. Rows correspond to the genes in each gene list and columns to the mean expression value for the genes at a given time point in a given group (treatment). Genes marked with an asterisk correspond to those found differentially expressed in at least one of the pairwise comparisons between the time points and groups. The gene-expression values were scaled by row to make them comparable over time and between groups. The boxplots for the expression of the genes of each gene list were produced using the *ggplot2* R package v 3.3.6 using the expression values as described in the heatmaps section. Mean expression values for each time point were statistically compared to assess their difference using the non-parametric Wilcoxon test and Benjamini-Hochberg correction for multiple testing, as implemented in the *ggpubr* R package v 0.4.0.

The evolution of the cell-type composition estimates based on the transcriptomic data of the PBMC samples was calculated from the bulk transcriptomic data using the Cibersort deconvolution method[28], as implemented in github.com/favilaco/deconv_benchmark. The method was applied to the LM22 cell-type signature, a leukocyte gene signature matrix containing 547 genes that distinguish 22 human hematopoietic cell phenotypes, including seven T-cell types, naive and memory B cells, plasma cells, NK cells, and myeloid subsets[28]. ANOVA was performed with Tukey multi-comparison tests to compare the inferred proportions between time points.

### Luminex
The plasma concentrations of the analytes IFNγ, IL1RA, IL6, IL8, MCP1, sCD40L, and VEGF were measured using NHP cytokine/chemokine magnetic bead panel I (Merck). The plasma concentrations of TNFα, Perforin, and FasL were measured using NHP cytokine/chemokine magnetic bead panel II (Merck). Samples and plates were prepared following the manufacturer's instructions and the measurements were performed on a Magpix Instrument (Merck).

### T-cell activation assay
T-cell activation in response to overlapping peptides was measured using an intracellular cytokine staining assay[23,24]. Fresh blood samples were incubated for 6 h at 37 °C with peptide pools corresponding to 15-mer peptides covering the full sequence of LASV Josiah GP or NP, with an 11-mer overlap, in addition to anti-CD28 and anti-CD49d antibodies and brefeldin A. Staphylococcus enterotoxin A and PBS served as positive and negative controls. Samples were then treated with PBS-EDTA before staining with CD3 CD3 (557597, APC mouse anti-human, clone SP34-2, 14 μL), CD4 (560836, Alexa Fluor[R] 700 mouse anti-human, clone L200, 5 μL), and CD8 (560179, APC-H7 mouse anti-human, clone SK1, 5 μL) antibodies from BD biosciences. After fixation and permeabilization, cells were stained with IFNγ (559327, PE mouse

anti-human, clone B27, 20 μL) and TNFα (557647, PE-Cy™7 mouse anti-human, clone Mab11, 5 μL) antibodies from BD Biosciences, CD137 (130-119-886, VioBright FITC mouse anti-human, clone 4B4-1, 2 μL), and CD154 (130-113-609, VioBlue mouse anti-human, clone 5C8, 2 μL) antibodies from Miltenyi Biotec. Cells were analyzed by flow cytometry using a 10-color Gallios cytometer (Beckman Coulter) and the collected data analyzed using Kaluza v.2.1 software Beckman Coulter).

## Enzyme-linked immunosorbent assays

LASV-specific IgM and IgG antibody titers were determined by enzyme-linked immunosorbent assays[23,24]. For anti-LASV IgM, IgM μ-chain (SAB3700778, Sigma-Aldrich, 5 μg/mL) coated Maxisorp plates were incubated with lysates of LASV Josiah-infected Vero E6 cells before the addition of diluted plasma (1:100, 1:400, and 1:1600). Plates were then treated with mouse anti-LASV monoclonal antibodies (a kind gift of P. Jahrling, USAMRIID) and peroxidase-conjugated anti-mouse antibodies (SAB3701029, Sigma-Aldrich, 1:20,000). For anti-LASV-IgG, Polysorp plates were first coated with lysates of LASV Josiah-infected Vero E6 cells or recombinant LASV Josiah NP or GP (Zalgen) and then incubated with plasma dilutions (1:250, 1:1000, 1:4000, and 1:16,000). Plates were treated with peroxidase-conjugated antibodies against nonhuman primate IgG (SAB3700766, Sigma-Aldrich, 1:5000). MeV-specific IgG antibody titers were also determined. Maxisorp plates were coated with inactivated MeV antigens (PR-BA 102, Jena Biosciences) and then incubated with diluted plasma (1:250, 1:1000, 1:4000, and 1:16,000. Final staining was performed using peroxidase-conjugated antibodies against nonhuman primate IgG (SAB3700766, Sigma-Aldrich, 1:5000). All plates were revealed using tetramethylbenzidine and quantification was performed on a Tecan analyzer. The threshold of positivity was fixed at 2× mean of the negative controls + 1 standard deviation (SD) of the mean for LASV IgM and IgG and at the mean of the negative controls + 2 SDs for MeV IgG.

## Measles neutralization assays

The titers of MeV-neutralizing antibodies were quantified using a protocol adapted from Malczyk et al.[46]. Briefly, two-fold dilutions (from 4 to 8192) of plasma collected at day 0 prior to LASV challenge were incubated for 1 h at 37 °C with 250 plaque-forming units (pfu) of MeV-GFP, a measles Schwarz vaccine strain expressing GFP. The mixtures were then added to Vero NK cells seeded 4 h prior to the assays at $1 \times 10^4$ cells per well. Cells were incubated for 4 days at 37 °C 5% $CO_2$ then the positivity of each well was evaluated using a Leica epifluorescence microscope. The MeV-neutralization titer represents the highest dilution at which no GFP-expressing cells could be detected.

## Statistical analysis

Statistical analyses of the flow cytometry data were performed using Sigma Plot 14.5 (Systat Software Inc.). Data were analyzed by one-way or two-way analysis of variance (ANOVA) if the data set passed a normality test (Shapiro–Wilk) and the equal variance test (Brown-Forsythe). If not, a Kruskal–Wallis one-way ANOVA on ranks was used. All other statistical analyses were performed using GraphPad Prism 9.4.1. Data for continuous variables are expressed as individual points or as the mean ± standard error of the mean (SEM). A Student's t-test or one-way ANOVA with multiple comparisons was used to compare the means between groups of data passing the normality test (Shapiro-Wilk). A Kruskal–Wallis multiple comparisons test was used to compare the means between groups of data with a non-normal distribution.

## Reporting summary

Further information on research design is available in the Nature Portfolio Reporting Summary linked to this article.

## Data availability

The RNA-seq data generated in this study have been deposited in the Zenodo database under accession code 7547502 and in NCBI's Gene Expression Omnibus under GEO accession number GSE225258. The primary data generated in this study are provided in the Source Data file. Biological materials may be available under reasonable request to the corresponding author. As it is protected by the U.S. patent 20200308555, the MeV-NP vaccine can only be obtained from authors for non-commercial research under a material transfer agreement. Source data are provided with this paper.

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

## Acknowledgements

We thank S. Mundweiller, S. Godard, E. Moissonnier, D. Thomas, S. Mely, B. Labrosse, D. Pannetier, and C. Leculier (P4 INSERM–Jean Merieux, US003, INSERM) for assistance in conducting the BSL-4 experiments. We are grateful to S. Becker for providing us with LASV strain Josiah. We also thank T.G. Ksiasek, P.E. Rollin, and P. Jahrling for the LASV monoclonal antibodies and L. Branco (Zalgen Labs) for providing recombinant LASV Josiah proteins. We are grateful to Themis Bioscience GmbH, a subsidiary of Merck & Co., Inc., Rahway, New Jersey USA (E. Tauber, A. Kort, K. Ramsauer, S. Schrauf, Y. Tomberger, and R. Tschismarov), the Coalition for Epidemic Preparedness and Innovations (R. Hatchett, G. Thiry, and M. Saville), and C. Gerke (Department of Innovation Development, Institut Pasteur) for invaluable support. This study was funded by a grant from the Coalition for Epidemic Preparedness and Innovations (CEPI-CfP-001) to S. Baize and by a grant from the Agence Nationale de la Recherche (ANR-21-CE18-0004-01) to M. Mateo.

## Author contributions

M.M. and S. Bai conceived and supervised the project. M.M. produced the MeV-NP vaccine. P.R. and L.F. took care of the monkeys and performed the immunizations of the D-16 animals. M.M., S.R., A.J., C.G., X.C., J.H., C.P., and V.B.-C. performed the RNA extractions, PBMC isolation. and flow cytometry sample preparation. A.V., S. Bar, A.D., O.J., O.L., and M.Di took care of the monkeys and performed the D-8 immunization, LASV challenge, scoring, sampling, hematological and biochemical analyses, and necropsies. C.C. and H.R. supervised and validated the BSL-4 protocols. M.M. and A.J. performed the ELISA. S.R. and S. Bai performed the flow cytometry analyses. M.M., A.J., and K.N. performed the quantitative PCR analyses. M.M. and K.N. performed the viral titrations and seroneutralization assays. M.M. and S.R. performed viral titrations on organs. M. Da and C.L.-L. generated the RNA-seq libraries and performed the sequencing. E.P., H.L-M., and N.P. analyzed the RNA-seq data. M.M. and S. Bai analyzed the data and wrote the paper, with the help of F.T.

## Competing interests

The authors declare that they have no competing interests. M.M., S.Bai, and F.T. are inventors of MeV-NP that is protected by US patent Lassa vaccine no. 20200308555.
