## [Peer Review File · Nature Communications]

Rapid protection induced by a single-shot Lassa vaccine in cynomolgus monkeysREVIEWER COMMENTS

Reviewer #1 (Remarks to the Author):

Figure 1 Panel C: Are the temperature charts supposed to be in the same order as the sub-panels in 1B, ie CT, D-16, D-8 and H+1? This doesn't appear to be the case as low temps and cessation of recording are evident in the 3rd and 4th panels from the left.

Figure 2 and Figure 6: Are these means, medians, and what are the error bars? If mean values, how do you know the data is normally distributed? Also, Figure 4: Data plotted are means + SEM. Have these data passed a normality test. The method section highlighted the statistical approach adopted and this appeared robust. However, it was less unclear which tests were applied where and what normality testing had been carried out. It would be helpful to add this detail to the individual legends.

Line 271: Given the number of animals it is tricky to apply statistical testing, therefore would it be more accurate to replace 'significant' with 'obvious' or 'major'.

Supp Fig 3: The legend states 2h – shouldn't this be +1h?

Under weaknesses I think it would be prudent of the authors to discuss what appears to be a relatively short window of protection. Their current and previous data shows that the -8 day vaccination is better than -16d and -1m. How quickly is protection lost – this is an important aspect for the authors to discuss.

Reviewer #2 (Remarks to the Author):

Dear Authors,

This manuscript evaluates the efficacy of a modified Measles virus (MV) vaccine to deliver Lassa Virus (LASV) structural nucleoprotein and glycoprotein genes for the induction of immunity to lethal LASV challenge. The primary hypothesis that was tested was whether the MV-NP vaccine can induce immunity rapidly. The study tested macaques that were vaccinated 16 and 8 days before challenge and 1 after challenge. The main takeaway from this study was that protection could be achieved by 8 days post-immunization. However, there were some indications that 8 days provided better protection than 16 days which may indicate a rapid waning of the immunity. Comparisons were made to previous studies that showed long-term (1 year) protection using this same vaccine, however it appears that these vaccinated macaques were not preimmune to MV and may have a very different immune profile and kinetics. In addition the MV vaccine was MV-CHIK which carries the Chikungunya structural genes and is not used for standard vaccination as it is still in clinical trials. The CHIK genes may have also skewed or disrupted the anti-MV immune responses. Overall the manuscript is complete and the analyses were comprehensive. The groups were small (n = 3), but that is often time part of the compromise with NHP studies. In general the studies were well designed and the results will contribute to the field of vaccine studies.

I have several suggestions that I believe will improve the manuscript and make it more reader friendly.

1) I think it's important to describe the context of pre-existing MV immunity earlier on in the manuscript. This is important since most people receive the MMR vaccine in early childhood and would have pre-existing immunity to MV. Also, there should be some discussion as to why the MV-CHIK vaccine was used as it does not accurately reflect the MV vaccine used in the field.

- 2) The MV-NP vaccine expresses both NP and GP, it seems appropriate to include an indication of both, such as MV-NG or MV-NP-GP.
- 3) I couldn't find sup Table 1.
- 4) Please expand on what appears to be rapidly waning immunity in the Day 16 group. When using the previous studies, the author should clarify how comparable the studies are to each other. For example were the previous studies done in naïve animals or did they receive the same MV-CHIK 1 year before the studies?
- 5) I would refer to the D16 and D8 groups as prophylactic and the H+1 as therapeutic, since it was given post-exposure.
- 6) Fig. 1 needs labels. Fig 1C does not match up with the figure legends in Fig 1B and it is assumed that the top label in Fig 1D represent all of the figures in the column. Include legends for all figures or group them into boxes.
- 7) When data averages are shown the appropriate statistics should be applied and reported. In addition statistical methods should be reported in the figure legend. For example in Fig 2 there are asterix, but no description of the statistical significance.
- 8) For ease of interpretation, it would be good to show a separate figure showing the pre-existing immunity (neutralizing antibody and not ELISA binding) to MV rather than trying to pull the data from Fig 4E
- 9) The data in Fig 7 is so complex it has lost much of its value. Meaning that if the data is so small that it can't be interpreted than it does not strengthen the manuscript. This data could be limited to the relative data that is described in the discussion.

Reviewer #3 (Remarks to the Author):

Mateo et al. describe non-human primate (NHP) studies on a lassa virus (LASV) vaccine candidate, MeV-NP, which was developed previously, tested in NHPs previously for its ability to protect against LASV challenge and is currently being evaluated in a phase 1 clinical trial.

In the current manuscript the authors test the same vaccine in NHPs in a different challenge schedule to investigate whether the vaccine can still protect when given shortly before the viral challenge. The underlying rationale is that a vaccine that generates protective immunity rapidly after one administration, would have value in an outbreak setting.

The results show that the vaccine protects NHPs from LASV challenge when given 8 days prior to challenge, but not when given an hour after challenge. Furthermore, the protection was stronger when given only 8 days prior to challenge compared with 16 days prior to challenge.

The study is solid and the experiments are performed to a high standard. Analysis of innate, T cell and B cell responses revealed that the cellular responses peaked around day 9, explaining the strong protection against challenge at day 8.

Overall this is a solid study with relevance for the deployment of this LASV vaccine, if and when it proves immunogenic and effective in humans. However, the study does not involve substantial scientific innovation and would therefore probably be more appropriate for another journal.

Reviewer #1 (Remarks to the Author):

We thanks the reviewer for her/his comments that we have addressed in this rebuttal and in the new version of the manuscript.

Figure 1 Panel C: Are the temperature charts supposed to be in the same order as the sub-panels in 1B, ie CT, D-16, D-8 and H+1? This doesn't appear to be the case as low temps and cessation of recording are evident in the 3rd and 4th panels from the left.

We are sorry for the confusion. Indeed the subpanels in Panel C are not in the same order as in panel B and D. We reorganized the figure and added a legend for more clarity.

Figure 2 and Figure 6: Are these means, medians, and what are the error bars? If mean values, how do you know the data is normally distributed? Also, Figure 4: Data plotted are means + SEM. Have these data passed a normality test. The method section highlighted the statistical approach adopted and this appeared robust. However, it was less unclear which tests were applied where and what normality testing had been carried out. It would be helpful to add this detail to the individual legends.

We are sorry for the lack of clarity regarding the different statistical analyses. We now provide information about normality and statistical tests in the individual legends as suggested by the reviewer for the figures 2, 4, and 5. However, as no statistics were performed on figure 6, the legend of this one has not been modified.

Line 271: Given the number of animals it is tricky to apply statistical testing, therefore would it be more accurate to replace 'significant' with 'obvious' or 'major'.

We replaced significant by obvious line 277.

Supp Fig 3: The legend states 2h – shouldn't this be +1h?

The figure was mislabeled. We replaced 2h by H+1.

Under weaknesses I think it would be prudent of the authors to discuss what appears to be a relatively short window of protection. Their current and previous data shows that the -8 day vaccination is better than -16d and -1m. How quickly is protection lost – this is an important aspect for the authors to discuss.

We understand the reviewer's comment but we do not agree with the statement about the "short window of protection". The vaccine protects at D-16 and D-8. We previously demonstrated that it also protects as efficiently after a single immunization one year or one month before the challenge. In this sense, we cannot say that the vaccine has a short window of protection but rather a large window of protection from 8 days up to a year. The "apparently better efficacy" (as stated line 431) of the D-8 regimen is already discussed in the manuscript. We have qualified our statement on the better efficacy of the D-8 regimen in the discussion, now stating that the differences were not significant between the D-16 and D-8 groups.

Reviewer #2 (Remarks to the Author):

Dear Authors,

We thank the reviewer for the helpful suggestions that helped improving the manuscript. We have addressed all comments in the rebuttal and in the new version of the manuscript. In addition, we now provide a new figure with measles seroneutralization titers before challenge to support the pre-existing immunity.

This manuscript evaluates the efficacy of a modified Measles virus (MV) vaccine to deliver Lassa Virus (LASV) structural nucleoprotein and glycoprotein genes for the induction of immunity to lethal LASV challenge. The primary hypothesis that was tested was whether the MV-NP vaccine can induce immunity rapidly. The study tested macaques that were vaccinated 16 and 8 days before challenge and 1 after challenge. The main takeaway from this study was that protection could be achieved by 8 days post-immunization. However, there were some indications that 8 days provided better protection than 16 days which may indicate a rapid waning of the immunity.

We do not agree on the “rapid waning of immunity”. Indeed, we already demonstrated that a single immunization of MeV-NP one year and one month before challenge was fully protective. The vaccine is similarly efficient 16 days before challenge. Although it seems to protect more efficiently at 8 days before challenge, the differences between the D-16 and D-8 groups are not significant, as we are now stating in the discussion. Altogether, there is no rapid waning of immunity and the vaccine provides a complete protection against LASV from 8 days to one year after immunization.

Comparisons were made to previous studies that showed long-term (1 year) protection using this same vaccine, however it appears that these vaccinated macaques were not preimmune to MV and may have a very different immune profile and kinetics.

We agree with the reviewer and we have modified this part of the discussion, now comparing more in details the different protocols (lines 417 to 430).

In addition the MV vaccine was MV-CHIK which carries the Chikungunya structural genes and is not used for standard vaccination as it is still in clinical trials. The CHIK genes may have also skewed or disrupted the anti-MV immune responses.

The MV vaccine used in this study was not the MV-CHIK but a classical measles live attenuated vaccine produced by the Serum Institute of India. We are sorry for the confusion and we now provide more information about the vaccine in the materials and methods section.

Overall the manuscript is complete and the analyses were comprehensive. The groups were small (n = 3), but that is often time part of the compromise with NHP studies. In general the studies were well designed and the results will contribute to the field of vaccine studies.

I have several suggestions that I believe will improve the manuscript and make it more reader friendly.

1) I think it's important to describe the context of pre-existing MV immunity earlier on in the manuscript. This is important since most people receive the MMR vaccine in early childhood and

would have pre-existing immunity to MV. Also, there should be some discussion as to why the MV-CHIK vaccine was used as it does not accurately reflect the MV vaccine used in the field.

We have added a part on pre-existing immunity to MeV in the introduction (line 85 to 88) and a few words in the abstract. We do not discuss the use of MV-CHIK as this is a misunderstanding already addressed before in the rebuttal.

2) The MV-NP vaccine expresses both NP and GP, it seems appropriate to include an indication of both, such as MV-NG or MV-NP-GP.

As we have used the name MeV-NP for this vaccine in two previous publications, we prefer keeping it like this to avoid confusion. A description of the vaccine is given line 79 and 80 and therefore should not mislead the reader.

3) I couldn't find sup Table 1.

We forgot to upload the table S1 while submitting the manuscript. We apologize for that and we now provide the table S1.

4) Please expand on what appears to be rapidly waning immunity in the Day 16 group. When using the previous studies, the author should clarify how comparable the studies are to each other. For example were the previous studies done in naïve animals or did they receive the same MV-CHIK 1 year before the studies?

We do not agree on the "rapid waning of immunity". Indeed, we already demonstrated that MeV-NP could protect efficiently one year and one month before challenge after a single immunization. It protects as efficiently 16 days before challenge and seems to protect more efficiently at 8 days before challenge but as we are now stating in the new discussion, the differences between the D-16 and D-8 groups are not significant. We have also added a paragraph on the comparability of the different studies (one year, one month, 16 days, 8 days) and added the paragraph discussing pre-existing immunity in this new paragraph.

5) I would refer to the D16 and D8 groups as prophylactic and the H+1 as therapeutic, since it was given post-exposure.

We have replaced post-exposure by therapeutic for all cases.

6) Fig. 1 needs labels. Fig 1C does not match up with the figure legends in Fig 1B and it is assumed that the top label in Fig 1D represent all of the figures in the column. Include legends for all figures or group them into boxes.

We are sorry for the confusion. Indeed the subpanels in Panel C are not in the same order as in panel B and D. We reorganized the figure and added a legend for more clarity.

7) When data averages are shown the appropriate statistics should be applied and reported. In addition statistical methods should be reported in the figure legend. For example in Fig 2 there are asterix, but no description of the statistical significance.

We are sorry for the lack of clarity regarding the different statistical analyses. We now provide information about statistical tests in the individual legends of figure 2, 4 and 5 as suggested by the reviewer and we added description of the statistical significance for figure 2.

8) For ease of interpretation, it would be good to show a separate figure showing the pre-existing immunity (neutralizing antibody and not ELISA binding) to MV rather than trying to pull the data from Fig 4E.

We have performed MeV seroneutralization assays to determine the titers of neutralizing antibody against MeV at the time of challenge. The new data is presented a new panel F in Figure 4. We modified the text in the results (lines 213 to 216), discussion (lines 534-535) and materials and methods (lines 669 to 676) sections accordingly.

9) The data in Fig 7 is so complex it has lost much of its value. Meaning that if the data is so small that it can't be interpreted than it does not strengthen the manuscript. This data could be limited to the relative data that is described in the discussion.

We agree that the figure is quite complex and rather small. As we believe the heatmaps are still informative, we decided to split this figure in two different figures, now Figure 7 and 8. The old Figure 8 is now Figure 9.

Reviewer #3 (Remarks to the Author):

We thank the reviewer for the positive review of our manuscript. We strongly believe that the manuscript will be of great interest for the readers of Nature Communications, especially considering that the results of the phase I clinical trials should be released shortly.

Mateo et al. describe non-human primate (NHP) studies on a lassa virus (LASV) vaccine candidate, MeV-NP, which was developed previously, tested in NHPs previously for its ability to protect against LASV challenge and is currently being evaluated in a phase 1 clinical trial.

In the current manuscript the authors test the same vaccine in NHPs in a different challenge schedule to investigate whether the vaccine can still protect when given shortly before the viral challenge. The underlying rationale is that a vaccine that generates protective immunity rapidly after one administration, would have value in an outbreak setting.

The results show that the vaccine protects NHPs from LASV challenge when given 8 days prior to challenge, but not when given an hour after challenge. Furthermore, the protection was stronger when given only 8 days prior to challenge compared with 16 days prior to challenge.

The study is solid and the experiments are performed to a high standard. Analysis of innate, T cell and B cell responses revealed that the cellular responses peaked around day 9, explaining the strong protection against challenge at day 8.

Overall this is a solid study with relevance for the deployment of this LASV vaccine, if and when it

proves immunogenic and effective in humans. However, the study does not involve substantial scientific innovation and would therefore probably be more appropriate for another journal.

REVIEWERS' COMMENTS

Reviewer #2 (Remarks to the Author):

Dear Authors,

Thank you for your revisions. I am satisfied with the manuscript. I believe this research is significant and will have an impact on science.